# Redefining the Task of Bioactivity Prediction

**Yanwen Huang[2,1][*][†], Bowen Gao[1,3][*], Yinjun Jia[1], Hongbo Ma[3],
Wei-Ying Ma[1], Ya-Qin Zhang[1], Yanyan Lan[1,4][‡]**

[1]Institute for AI Industry Research (AIR), Tsinghua University
[2]Department of Pharmaceutical Science, Peking University
[3]Department of Computer Science and Technology, Tsinghua University
[4]Beijing Academy of Artificial Intelligence (BAAI)

## Abstract

Small molecules are vital to modern medicine, and accurately predicting their bioactivity against protein targets is crucial for therapeutic discovery and development. However, current machine learning models often rely on spurious features, leading to biased outcomes. Notably, a simple pocket-only baseline can achieve results comparable to, and sometimes better than, more complex models that incorporate both the protein pockets and the small molecules. This phenomenon arises from insufficient training data and an improper evaluation process, which is typically conducted at the pocket level rather than the small molecule level. To address these issues, we redefine the bioactivity prediction task by introducing the **SIU** dataset-a million-scale **S**tructural small molecule-protein **I**nteraction dataset for **U**nbiased bioactivity prediction task, which is 50 times larger than the widely used PDBbind. The bioactivity labels in SIU are derived from wet experiments and organized by label types, ensuring greater accuracy and comparability. The complexes in SIU are constructed using a majority vote from three commonly used docking software programs, enhancing their reliability. Additionally, the structure of SIU allows for multiple small molecules to be associated with each protein pocket, enabling the redefinition of evaluation metrics like Pearson and Spearman correlations across different small molecules targeting the same protein pocket. Experimental results demonstrate that this new task provides a more challenging and meaningful benchmark for training and evaluating bioactivity prediction models, ultimately offering a more robust assessment of model performance. Dataset and Code are available at: https://github.com/bowen-gao/SIU.

## 1 Introduction

Small molecules are essential active components in life-saving therapeutic drugs, with their safety and efficacy intricately linked to interactions with various protein targets within the human body. Consequently, bioactivity prediction is a critical task in the drug discovery process (Tropsha et al., 2024; Gaulton & Overington, 2010), driven by the rapid advancement of machine learning methods. In this context, "bioactivity" encompasses the diverse biological effects resulting from small molecule-protein interactions, including binding responses-commonly quantified by the dissociation constant $(K_d)$ and the inhibition constant $(K_i)$-as well as functional responses, typically assessed through the half-maximal inhibitory concentration $(IC_{50})$ and the half-maximal effective concentration $(EC_{50})$.

Recently, various 3D machine learning models have been proposed in this direction (Townshend et al., 2020; Zhou et al., 2022; Gao et al., 2023a; Luo et al., 2023), achieving significant advancements. These methods utilize the structural information of small molecules and protein targets as inputs to learn a mapping function between these inputs and bioactivity labels. This methodology is inherently sound and explainable, as biological insights suggest that the biological effect of a small

---

[*]Equal contirbution
[†]Work was done while Yanwen Huang was an intern at AIR.
[‡]Correspondence to lanyanyan@air.tsinghua.edu.cn

molecule largely depends on its 3D shape complementarity with its protein targets (Verma et al., 2010), a principle known as the key-lock modulation theory (Koshland Jr, 1995; Eschenmoser, 1995). Nevertheless, the applications of these methods have not yielded satisfactory results regarding drug discovery capabilities. For instance, when using predicted biological labels to differentiate between active and inactive molecules-an essential task in virtual screening-these predictive models often fail to compete with widely used docking methods, as noted in Shen et al. (2021) and Gao et al. (2023b).

Our analysis reveals that these models can be easily biased to some spurious features, leading to inaccurate predictions based on shortcuts. We propose a pocket-only baseline to diagnose the current bioactivity prediction task. While previous works assume that the bioactivity labels are determined by the interaction between small molecules and protein targets, they tend to assess only the protein target while ignoring the provided small molecules, representing a degenerate solution. As shown in Figure 1A and 1B, experiments on the widely used Atom3D ligand binding affinity (LBA) prediction dataset (Townshend et al., 2020) demonstrate that this pocket-only approach achieves, or can even outperforms, models utilizing the complex information across both 30% and 60% sequence identity splits. These results support our claim by suggesting that statistical irregularities in the data enable a model to achieve bioactivity predictions beyond what should be possible without access to the small molecule information.

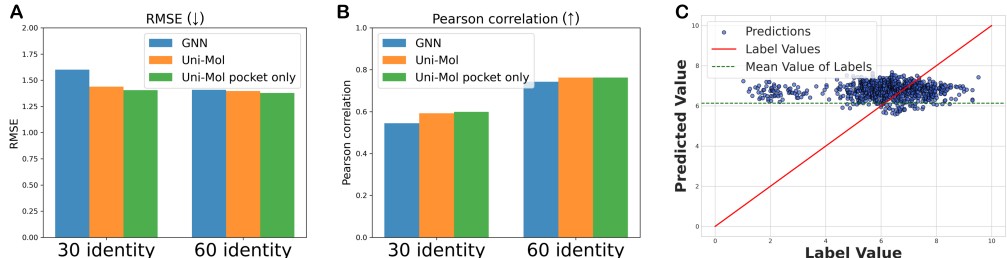

Figure 1: **Analysis of Atom3D bioactivity prediction task.** The evaluation metrics include **(A)** Root Mean Squared Error (RMSE) and **(B)** Pearson correlation. The models tested include: a GNN model using the full protein-ligand complexes as inputs, a Uni-Mol model with both a small molecule encoder and a protein encoder, and a Uni-Mol model with only a protein encoder which only takes pocket side information. Performance is evaluated across different sequence identity splits (30% and 60%). It shows that **Pocket-only model can overfit Atom3D bioactivity prediction task. (C)** Predicted versus actual label values for various small molecules within a single protein target.

Upon further analysis, we find that the key issue stems from the improper definition of the current bioactivity prediction task, particularly in terms of both data construction and evaluation metrics.

From a data perspective, the constructed training data is not sufficient for developing a robust bioactivity predictor. Although previous works have utilized different training data, they are all derived form PDBbind (Wang et al., 2004; 2005), which contains only about 20,000 small molecule-protein target pairs. More importantly, for each protein target, these datasets typically feature only a single small-molecule ligand. This introduces bias into the training data, causing models to primarily learn the bioactivity range for each protein target rather than differentiating between various small molecules interacting with the same target. As demonstrated in Figure 1C, when testing a model with different small molecules for the same target, even with both protein and small molecule information provided, the model generates predictions that cluster around the mean bioactivity value of the target, while the actual label values vary across a much wider range. This behavior suggests that the model trained on the current dataset fails to differentiate between different small molecules. This also helps to explain why pocket-only baselines can achieve unexpectedly good metric values.

From an evaluation perspective, the current metrics fail to accurately reflect how well models capture the interactions between a protein target and diverse small molecules. Specifically, established metrics like Pearson and Spearman correlations are computed across different protein targets rather than across multiple small molecules for the same target. This approach primarily measures differences between various protein targets. Consequently, models can overfit by relying predominantly on pocket information without truly learning the nuances of small molecule binding.

To address these issues, we propose redefining the bioactivity prediction task in this paper. Our strategy involves constructing a novel, large-scale structural dataset of small molecule-protein interactions, featuring multiple small molecules for each protein target, and evaluating metrics across these different small molecules. A significant challenge lies in constructing a large-scale dataset of reliable small molecule-protein complexes, as high-quality structural data depends on labor-intensive and time-consuming wet-lab experiments. To tackle this, we first sourced, cleaned, and deduplicated small molecules and protein targets from relevant databases containing high-quality bioactivity labels. For each protein and its various pockets, we utilized multiple docking software programs, such as Vina (Trott & Olson, 2010), to dock associated molecules, generating primary interaction complex structures. Subsequently, a majority vote mechanism was employed to obtain high-quality interaction poses. Furthermore, we differentiated between various label types, such as $K_d$, $K_i$, $IC_{50}$, and $EC_{50}$, to mitigate potential biases associated with label types during training and evaluation. This resulted in a large-scale **S**tructural dataset of small molecule-protein **I**nteractions for **U**nbiased bioactivity prediction, namely **SIU**.

The SIU dataset comprises over 5.34 million conformations and features 1.38 million rigorously curated bioactivity annotations, each clearly designated by label types. This extensive dataset provides comprehensive coverage of diverse small molecules, surpassing the limitations of datasets restricted to molecules structurally similar to co-crystal ligands. It also includes a wide array of protein targets across all major protein classes, with each protein linked to multiple PDB IDs that reflect distinct pocket conformations (not necessarily different binding sites). Notably, SIU differs from existing datasets that often overlook critical distinctions between label types, making it more suitable for fair bioactivity prediction and comparison.

With the availability of multiple small molecules with bioactivity labels for each protein target in SIU, we redefine the evaluation metrics by calculating values among different small-molecule ligands with the same target, rather than across different targets. The results are then averaged across targets using mean pooling. This approach ensures that the evaluation metrics accurately reflect the biactivity difference between small molecules within the same targets, thereby mitigating the aforementioned evaluation bias.

We compare the experimental results of training several classical baseline models on PDBbind and SIU. Two key findings highlight the outperformance of SIU over PDBbind. First, when evaluated using traditional metrics like RMSE, Pearson, and Spearman correlations across different targets, models trained on SIU demonstrate significant improvements compared to those trained on PDBbind, reflecting the value of the inclusion of more structural data. Notably, this performance enhancement persists even after removing data with high sequence identity from the test set, while models trained on PDBbind do not undergo the same removal. Second, our redefined metrics reveal a substantial drop in performance when evaluating small molecules within the same target. For instance, the Pearson correlation for $K_i$ can decrease from 0.485 to 0.036. This indicates that the new task is more challenging and that the bioactivity prediction abilities of the previous models may be overestimated due to improper task definitions. These results underscores the importance of the unbiased bioactivity prediction task we introduced, which we believe will advance the development of machine learning models that are truly beneficial for drug discovery.

## 2 RELATED WORK

Commonly used bioactivity prediction tasks include the Comparative Assessment of Scoring Functions (CASF) task (Cheng et al., 2009; Li et al., 2014b;a; Su et al., 2018) and the Atom3D LBA task (Townshend et al., 2020). Both tasks are derived from the PDBbind dataset (Wang et al., 2004; 2005), which is widely used and contains complex structures of small molecule-protein interactions along with their corresponding bioactivity labels. However, the data cleaning and splitting methods differ between these tasks. The CASF-2016 task (Su et al., 2018) consists of 285 protein-ligand complexes, each labeled with an experimentally measured binding affinity. Since it does not provide a dedicated training set, prior research typically relies on self-defined training datasets derived from PDBbind. In contrast, the LBA task in Atom3D (Townshend et al., 2020) provides predefined training and testing splits, using sequence identity-based splits on 30% and 60% to ensure that test results reflect the model's generalization ability. This task combines different label types, including $IC_{50}$, $K_i$, and $K_d$, into a unified prediction variable, with a total of 4,463 complexes in the dataset.

In this work, we introduce the SIU dataset to address specific challenges in bioactivity prediction tasks. Similarly, large-scale, high-quality datasets like Papyrus (Béquignon et al., 2023), curated from diverse sources, address other critical aspects and contribute valuable resources to the field.

Atom3D also introduced two widely adopted baseline models: a voxel-grid-based 3D convolutional neural network (3D-CNN) and a graph neural network (GNN) (Townshend et al., 2020). Recent advances in bioactivity prediction have been driven by the application of pretrained models, such as Uni-Mol (Zhou et al., 2022) and ProFSA (Gao et al., 2023a). These models utilize large-scale pretraining on molecular and structural data to achieve state-of-the-art performance across various bioactivity prediction tasks. In Atom3D, binding affinity prediction models are evaluated using RMSE, Mean Absolute Error (MAE), Pearson correlation, and Spearman correlation metrics.

# 3 METHODS

## 3.1 SIU DATASET CONSTRUCTION

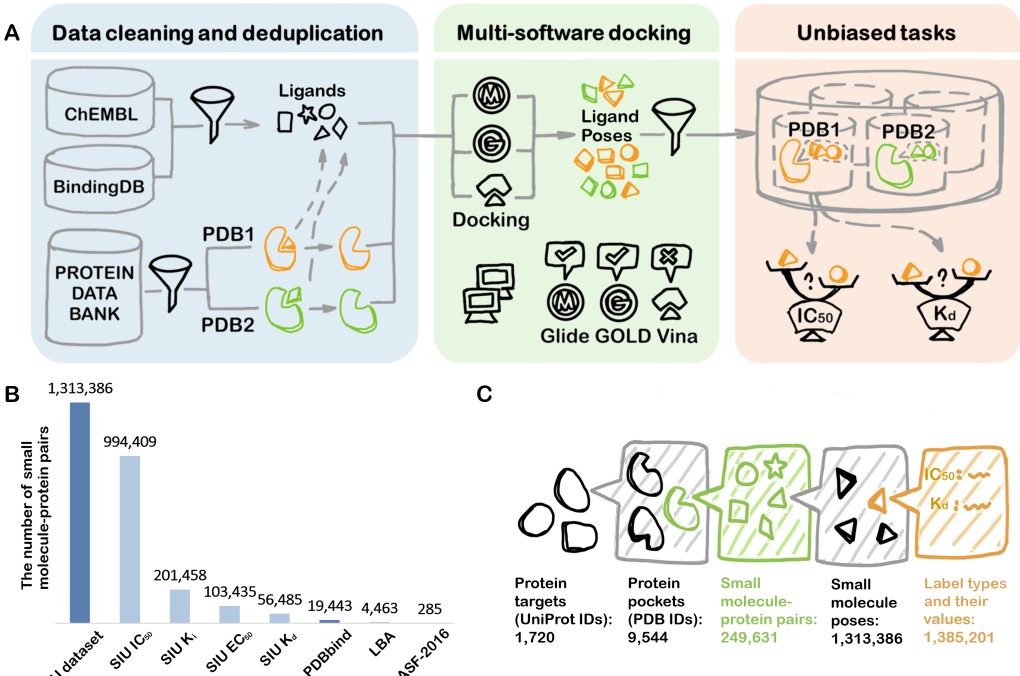

Figure 2: **Construction and features of the SIU dataset. (A)** The construction pipeline began with the collection of small molecules and protein targets from established databases, followed by data cleaning and deduplication. The small molecules underwent a comprehensive multi-software docking process, where they were prepared and docked to their experimentally validated targets. For quality control, the resulting poses were filtered through a voting mechanism, resulting in a dataset organized by both PDB and assay, designed to enable unbiased bioactivity prediction. **(B)** The SIU dataset offers large-scale structural data, making it more than fifty times the size of PDBbind and significantly larger than datasets currently used for bioactivity prediction tasks. **(C)** The SIU dataset is meticulously structured to enhance unbiased bioactivity prediction. It features multiple pockets (identified by PDB IDs) associated with the same protein target, multiple small molecules mapped to individual pockets (green), multiple high-quality docking poses per small molecule, and detailed label type annotations corresponding to all bioactivity values (orange).

**Bioactivity label data cleaning and deduplication.** Non-structural bioactivity data were retrieved from ChEMBL (Mendez et al., 2019; Gaulton et al., 2012) and BindingDB (Chen et al., 2001; Liu et al., 2007; Gilson et al., 2016). Non-drug-like small molecules were excluded based on criteria such as molecular weight (150–650 Da), the presence of at least one carbon atom, and a

minimum of nine heavy atoms (details in Appendix C.1). Each small molecule retained its original IUPAC International Chemical Identifier (InChI) keys (Heller et al., 2015) and Simplified Molecular Input Line Entry System (SMILES) notations (Weininger, 1988; Weininger et al., 1989) to prevent mismatches arising from different software calculations. Small molecules were deduplicated using Extended-Connectivity Fingerprints (ECFP) (Rogers & Hahn, 2010). Molecules with a Tanimoto similarity greater than 0.8 were clustered, and representatives were selected based on bioactivity, ensuring both quality and structural diversity while reducing computational expense in molecular docking. Deduplication was applied only to protein targets with a small molecule count exceeding 2,146, the 90th percentile across all targets.

Protein target information for each assay was standardized using UniProt IDs (Consortium, 2015; uni, 2017), ensuring consistency across datasets and alignment with structural data. Protein structures were retrieved, and pockets were extracted. An area within a 15 Å radius of the co-crystal ligand in the same complex structure is defined as a distinct pocket (identified by a single PDB ID), even if it belongs to the same binding site as pockets from other PDB files of the same protein. A filtering mechanism excluded PDB files with non-specific or irrelevant ligands, and pockets were further deduplicated using Fast Local Alignment of Protein Pockets (FLAPP). Bioactivity labels were standardized to molar units ($mol/L$) and converted to their negative logarithms, following conventions for drug-target binding affinity datasets (Öztürk et al., 2018). The resulting dataset, featuring structural pocket information, non-structural small molecule SMILES, and bioactivity labels.

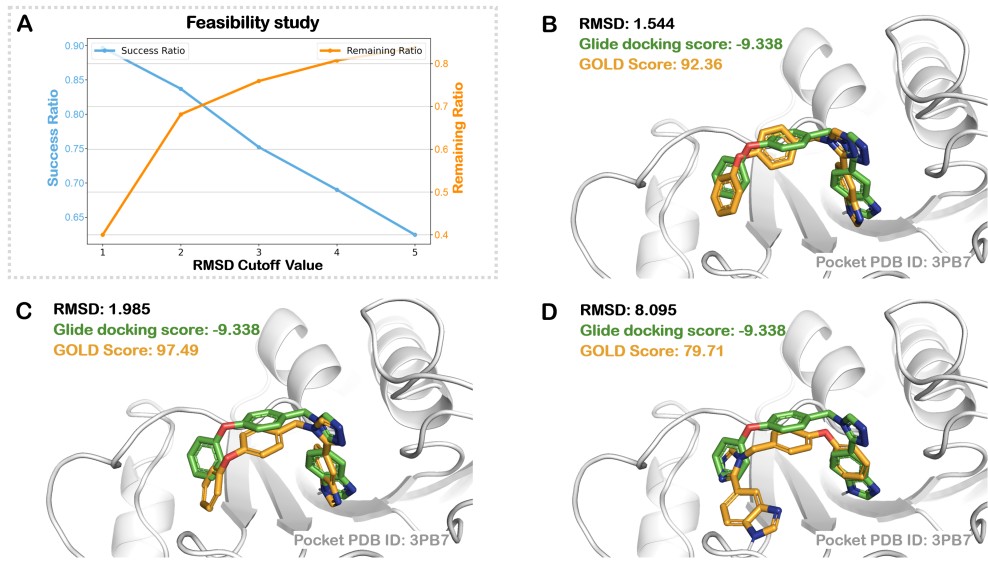

Figure 3: **Quality control of SIU structural data.** **(A)** A feasibility study of our methods showing the impact of root mean square deviation (RMSD) on success (when the pose simultaneously passes the consensus filter and has an RMSD < 2 Å compared to the co-crystal pose) and remaining ratios (the ratio of poses passing the filter) was analyzed using co-crystal poses, treated as the ground truth, and redocked into their original PDB pockets according to our docking procedure. **(B-D)** Visualization of our pose consensus mechanism, where RMSD is calculated between different docking poses from different software (within same pocket PDB ID 3PB7). A single Glide docking pose is compared with the top three docking poses generated by GOLD. **(B)** RMSD 1.544 Å: well-superimposed poses; **(C)** RMSD 1.985 Å: similar predicted binding modes; **(D)** RMSD 8.095 Å: fundamentally different predicted binding modes.

**Structural data construction via multi-software docking**  SIU employs multiple docking software programs (Friesner et al., 2004; Verdonk et al., 2003; Trott & Olson, 2010), reducing reliance on any individual docking software. Initial 3D conformations for the small molecules were generated prior to docking using the Glide LigPrep module with default settings. The preprocessed data were organized into formats compatible with the chosen docking software. Protein targets were prepared, and grid

files were generated according to each software's specific requirements to ensure compatibility. Small molecules were then docked into the pockets of the protein structures (detailed in Appendix C.2).

For quality control, the SIU structural data underwent a majority voting mechanism: only docking poses consistent across at least two of the three docking software were retained. This consensus-based approach mitigated the inclusion of erroneous or misleading docking poses, thereby improving the overall quality and reliability of the dataset.

We investigated the selection of the consensus filtering RMSD cutoff by evaluating the trade-off between pose accuracy and the quantity of retained data. Experiments were conducted to assess the impact of varying RMSD cutoffs on these factors (Figure 3A). In this experiment, we re-docked small-molecule ligands with known co-crystal structures using different docking software. A successful docking pose was defined as one with an RMSD of less than 2 Å compared to the experimental structure. The results demonstrate that with an RMSD cutoff below 2 Å, a significant number of molecules are retained, and the success rate of the poses is satisfactory. However, as the RMSD cutoff increases, the number of retained poses rises slightly, but their accuracy decreases substantially. This suggests that our consensus method is effective for quality control of docked structures. Furthermore, Figure 3B-D show that when the RMSD is around 2 Å, key interactions are preserved, indicating a potentially valid docking result. Based on these observations, an RMSD cutoff of 2 Å was selected as the optimal threshold.

## 3.2 DATASET OVERVIEW

**Large-scale.** The SIU dataset comprises 5,342,250 conformations detailing small molecule-protein interactions, each entry providing comprehensive structural and bioactivity information, as shown in figure 2B. It includes 1,385,201 bioactivity labels derived from wet experiments, each with standardized values and clearly annotated label types. The top four label types by small molecule-protein pair count are half-maximal inhibitory concentration $IC_{50}$ (994,409), $K_i$ (201,458), half-maximal effective concentration $EC_{50}$ (103,435), and $K_d$ (56,485), which form the primary subset used in our subsequent experiments.

**Diversity.** SIU offers an extensive range of data, encompassing 214,686 diverse small molecules and 1,720 distinct protein targets. It includes experimentally validated low-bioactivity or inactive molecules, which are often absent in structural datasets from wet experiments, thus providing valuable negative data for AI-driven drug discovery (AIDD). The dataset features broad protein type coverage, including proteins from different species and major protein classes. As illustrated in Figure 4D, the assay values of different protein targets vary significantly. This broad coverage ensures a comprehensive representation of small molecule-protein interaction modes, enhancing the relevance of our bioactivity prediction tasks to real biological environments.

**High-quality.** The structural information on small molecule-protein interactions in SIU is of high quality, due to our multi-software voting mechanism that maximizes docking accuracy within computational limits. As detailed in the structural data construction section, we achieved a satisfactory balance between data accuracy and scale, presenting high-quality data unobtainable with a single docking software or solely by ranking based on software-predicted docking scores. Docking software often provides successful simulated docking poses within the top-ranking positions, but these are not always ranked first by docking scores. Our method, however, is based on the consistency of docking pose sampling across different algorithms. By examining consensus among different docking algorithms, we effectively ensure more accurate docking pose data.

**Well-organized.** SIU's bioactivity labels are meticulously curated and systematically organized by PDB IDs and label types, ensuring data integrity and enabling effective PDB-wise and assay-wise comparisons. This organization offers a robust resource for unbiased bioactivity prediction, addressing the limitations of existing datasets that often fail to distinguish clearly between different bioactivity label types. Traditional machine learning measurements of correlations in bioactivity prediction tasks are often ineffective due to the lack of clarity in existing datasets. SIU can also address this problem, ensuring more precise and meaningful analyses. Our structured approach facilitates nuanced assessments, such as evaluating the impact of specific small molecule transformation on protein interactions or comparing the efficacy of different compounds within the same protein pocket context.

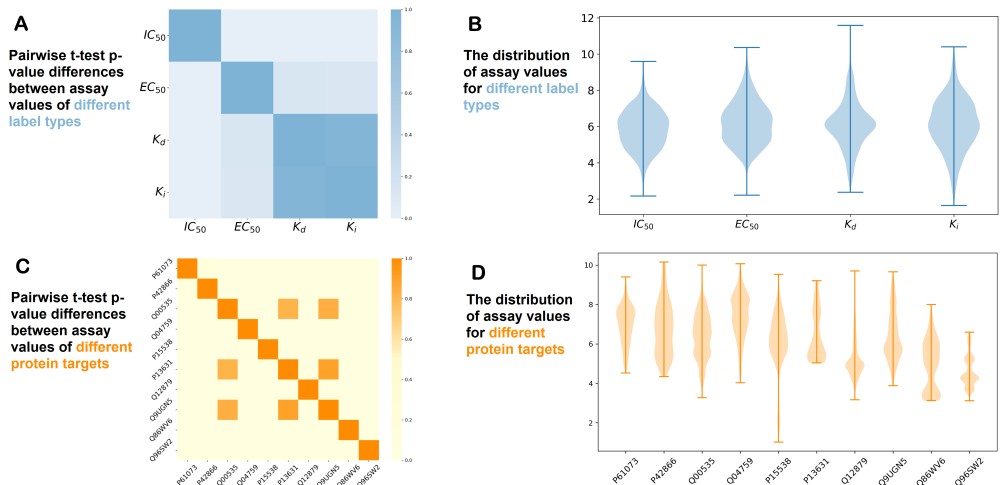

Figure 4: **Differences in assay values across label types and protein targets.** **(A)** The mean assay values vary among representative label types, as shown by a heatmap of pairwise t-test p-values. Smaller p-values (lighter colors) indicate significant differences. **(B)** Violin plots illustrate the distributions of label values for different label types. **(C)** Differences in mean assay values among ten protein targets. **(D)** The distributions of label values for different protein targets.

**Versatile-usage.** By providing a well-structured SIU dataset, we not only support unbiased bioactivity prediction but also enable a deeper understanding of the molecular mechanisms underlying these interactions. This level of detail is also crucial for training robust models for other small molecule-protein structure-based downstream tasks, such as protein-ligand docking, virtual screening, and molecular generation, as discussed in Appendix F. Additionally, we provide parallel data of small molecule-protein pairs that have not been subjected to docking, meaning they lack structural ligand poses but retain valuable data points that either could not generate a pose or did not meet quality control standards. This alternative dataset may be useful for evaluating bioactivity prediction models that do not rely on structural ligand poses, particularly when comparing methods that require 3D input data.

### 3.3 REFRAMING THE BIOACTIVITY PREDICTION TASK

**Organization of different label types.** We organized the data by label types to address the common issue of mixing $K_d$ and $K_i$ data, while also ensuring that other bioactivities are not neglected. As illustrated in Figure 4A and 4B, the physical meanings of the label types differ, leading to variations in their mean values and distributions. This highlights the importance of not mixing different label types and suggests that they should be treated as distinct tasks.

**Unbiased correlation metrics with group-by-pocket approach.** As shown in Figure 4C, we selected 10 different protein targets along with their corresponding $IC_{50}$ label values and calculated the pairwise t-test p-values. A higher p-value (darker color) indicates a higher similarity in the mean values of different targets. This observation is further corroborated by Figure 4D, where violin plots depict the distribution of label values for each of the representative targets, highlighting variations in both mean values and overall distributions. The figure clearly demonstrates that most target pairs exhibit significantly different distributions. These differences introduce bias into the dataset and explain why utilizing only pocket information can still achieve strong Pearson and Spearman correlation performance, as shown in Figure 1.

Thanks to the fact that our dataset provides multiple small-molecule ligands for each protein, we can reframe the task and introduce new bioactivity prediction metrics, enabling more unbiased benchmarking of the models.

In the traditional machine learning approach, Pearson correlation is calculated across all ligand-pocket pairs without considering the individual protein pocket. Given $N$ ligand-pocket pairs, where $\hat{y}_i$ represents the predicted bioactivity and $y_i$ represents the true bioactivity for each ligand-pocket

pairs, the Pearson correlation $r$ is computed as:

$$r = \frac{\sum_{i=1}^{N}(\hat{y}_i - \bar{\hat{y}})(y_i - \bar{y})}{\sqrt{\sum_{i=1}^{N}(\hat{y}_i - \bar{\hat{y}})^2}\sqrt{\sum_{i=1}^{N}(y_i - \bar{y})^2}}, \tag{1}$$

where: $\bar{\hat{y}}$ is the mean of the predicted bioactivities across all ligand-pocket pairs. $\bar{y}$ is the mean of the true bioactivities across all ligand-pocket pairs.

We calculate Pearson correlation after grouping by protein pockets (PDB IDs). For each protein pocket $t$, with $n_t$ ligands and their corresponding predicted bioactivities $\hat{y}_{i,t}$ and true bioactivities $y_{i,t}$, we first compute the Pearson correlation $r_t$ for each pocket:

$$r_t = \frac{\sum_{i=1}^{n_t}(\hat{y}_{i,t} - \bar{\hat{y}}_t)(y_{i,t} - \bar{y}_t)}{\sqrt{\sum_{i=1}^{n_t}(\hat{y}_{i,t} - \bar{\hat{y}}_t)^2}\sqrt{\sum_{i=1}^{n_t}(y_{i,t} - \bar{y}_t)^2}}, \tag{2}$$

where: $\bar{\hat{y}}_t$ is the mean of predicted bioactivities for ligands within the same pocket $t$. $\bar{y}_t$ is the mean of true bioactivities for ligands within the same pocket $t$.

Once we have computed the Pearson correlation for each pocket, the overall correlation considering all protein pockets (Pearson*) is obtained by mean pooling:

$$r^{\star} = \frac{1}{T}\sum_{t=1}^{T} r_t. \tag{3}$$

where $T$ is the total number of pockets.

The similar method is also applied to the Spearman correlation to get Spearman*.

This grouped-by-pocket approach offers an unbiased and more useful evaluation, as it ensures that the correlation reflects the model's ability to predict bioactivities for different small-molecule ligands within the same protein pocket, reducing bias introduced by variations between different pockets.

**Dataset Splits.** To ensure robust evaluation and flexibility, the dataset includes multiple predefined splitting strategies. These include sequence identity filters at thresholds of 90%, 60%, and 30%, as well as a combined sequence identity and structural similarity filter. A manually curated test set focuses on biologically meaningful tasks by incorporating representative protein targets across diverse classes, offering insights into the generalizability of predictions for key biochemically relevant targets. Additionally, bioactivity prediction models can be assessed using a 10-fold cross-validation framework, providing a reliable and unbiased approach for diverse training and testing scenarios (details in Appendix A.3).

## 4 EXPERIMENTS AND ANALYSIS

### 4.1 EXPERIMENTS

We conducted experiments using several classical models to provide baseline results and analyze our SIU dataset. The models tested include a voxel-grid based 3D-CNN model, a Graph Neural Network (GNN) model, and pretrained models such as Uni-Mol and ProFSA (Gao et al., 2023a) as it achieves SOTA result for protein-ligand binding affinity prediction task. Our experiments were performed in both Multi-Task Learning (MTL) and single-target settings. In the MTL setting, all data were combined to train a single MTL model. In the single-target setting, the Uni-Mol model was trained separately on individual labels.

The metrics used in our analysis include RMSE, MAE, general Pearson and Spearman correlation, and the correlation after grouping by PDB IDs. The general Pearson and Spearman correlations are calculated by mixing pairs of protein pockets and molecules. The grouped correlation metrics are calculated for different molecules within a single protein pocket. We use Pearson* to represent Pearson correlation after grouping by PDB ID (pocket), and Spearman* to represent Spearman correlation after grouping by PDB IDs.

Results for multi-task learning is shown in Table 1, and the results for single task learning is shown in Table 2.

Table 1: Results for multi task learning with different label types. We show results for 3D-CNN, GNN, Uni-Mol, and ProFSA trained on SIU 0.9 version.

| | | RMSE ↓ | MAE ↓ | Pearson ↑ | Pearson* ↑ | Spearman ↑ | Spearman* ↑ |
|---|---|---|---|---|---|---|---|
| $IC_{50}$ | 3D-CNN | 1.560 | 1.275 | 0.158 | 0.044 | 0.154 | 0.040 |
| | GNN | 1.412 | 1.141 | 0.336 | 0.241 | 0.316 | 0.235 |
| | Uni-Mol | **1.353** | **1.092** | **0.462** | **0.343** | **0.466** | **0.351** |
| | ProFSA | 1.361 | 1.108 | 0.382 | 0.331 | 0.356 | 0.317 |
| $EC_{50}$ | 3D-CNN | 1.518 | 1.234 | 0.128 | 0.010 | 0.128 | 0.004 |
| | GNN | 1.334 | 1.025 | **0.444** | 0.108 | 0.481 | 0.120 |
| | Uni-Mol | 1.273 | 1.017 | 0.428 | 0.178 | 0.461 | 0.144 |
| | ProFSA | **1.255** | **0.971** | 0.438 | **0.204** | **0.495** | **0.154** |
| $K_i$ | 3D-CNN | 1.534 | 1.260 | 0.201 | 0.025 | 0.200 | 0.021 |
| | GNN | 1.814 | 1.504 | 0.247 | 0.099 | 0.107 | 0.058 |
| | Uni-Mol | 1.390 | **1.133** | 0.375 | 0.092 | 0.324 | 0.056 |
| | ProFSA | **1.374** | 1.142 | **0.405** | **0.149** | **0.365** | **0.127** |
| $K_d$ | 3D-CNN | 1.503 | 1.233 | **0.173** | 0.024 | **0.167** | 0.038 |
| | GNN | 1.711 | 1.431 | -0.068 | 0.065 | -0.147 | 0.033 |
| | Uni-Mol | **1.429** | **1.223** | -0.084 | **0.155** | -0.175 | **0.144** |
| | ProFSA | 1.546 | 1.334 | -0.172 | 0.057 | -0.205 | 0.029 |

Table 2: Results for single task training with different label types. We show the results with Uni-Mol model on PDBbind dataset, our SIU 0.6 version and 0.9 version dataset.

| | Train Set | RMSE ↓ | MAE ↓ | Pearson ↑ | Pearson* ↑ | Spearman ↑ | Spearman* ↑ |
|---|---|---|---|---|---|---|---|
| $IC_{50}$ | PDBbind | 1.575 | 1.279 | 0.430 | 0.245 | 0.425 | 0.229 |
| | SIU 0.6 | 1.407 | 1.138 | 0.461 | 0.317 | 0.463 | 0.311 |
| | SIU 0.9 | **1.357** | **1.099** | **0.470** | **0.345** | **0.474** | **0.347** |
| $EC_{50}$ | SIU 0.6 | 1.400 | 1.163 | 0.280 | 0.171 | 0.284 | **0.150** |
| | SIU 0.9 | **1.340** | **1.096** | **0.384** | **0.196** | **0.379** | 0.142 |
| $K_i$ | PDBbind | 1.315 | 1.085 | 0.368 | 0.040 | 0.323 | 0.026 |
| | SIU 0.6 | 1.255 | 1.034 | 0.472 | **0.106** | 0.452 | **0.112** |
| | SIU 0.9 | **1.235** | **1.017** | **0.485** | 0.036 | 0.452 | 0.041 |
| $K_d$ | PDBbind | 1.565 | 1.308 | **0.041** | 0.010 | **0.004** | 0.006 |
| | SIU 0.6 | 1.389 | 1.192 | -0.149 | 0.052 | -0.206 | 0.022 |
| | SIU 0.9 | **1.364** | **1.141** | -0.033 | **0.103** | -0.082 | **0.065** |

## 4.2 ANALYSIS

**Different label types.** The bioactivity prediction difficulty varies among different label types. The $K_d$ task is the most challenging, primarily due to the varying correlations between different label types, as shown in Figure 4A and 4B. Although the means of $K_i$ and $K_d$ labels do not differ statistically, the distribution of these two data groups is different. The intrinsic differences in label types of bioactivity arise from the principles of the wet-lab experiments used to measure them. Binding assays focus on the direct interaction between the small molecule and the protein target, providing insights into the strength and specificity of this binding through metrics like $K_i$ and $K_d$, using techniques such as surface plasmon resonance (SPR) (Schasfoort, 2017; Englebienne et al., 2003) and isothermal titration calorimetry (ITC) (Leavitt & Freire, 2001). In contrast, functional assays measure the biological response elicited by the small molecule on the target, capturing its effect on a biological system and often quantified by $IC_{50}$ and $EC_{50}$ by enzyme activity assays (Bisswanger, 2014; Hall, 1996) or other wet experiment techniques. **The inherent differences in what these assays measure mean that their values cannot be directly compared (Yung-Chi & Prusoff, 1973).** Furthermore, even within the categories of binding and functional assays, metrics should not be used interchangeably, as $K_i$ and $K_d$ describe different aspects of binding affinity, just as $IC_{50}$ and $EC_{50}$ describe different aspects of biological response.

**Influence of our unbiased metrics** As demonstrated in Figure 5, calculating the correlation at after grouping by PDB ID (pocket) across all label types results in a significant decline in both Pearson and Spearman correlations. This observation suggests that it is more challenging to achieve high correlation when assessing binding affinities for different molecules within the same pocket after grouping. This challenge primarily arises from the different distribution of binding affinities across various protein pockets, as shown in Figure 4C and 4D. Furthermore, these findings highlight that **conventional machine learning approaches to measuring correlation without grouping by target**

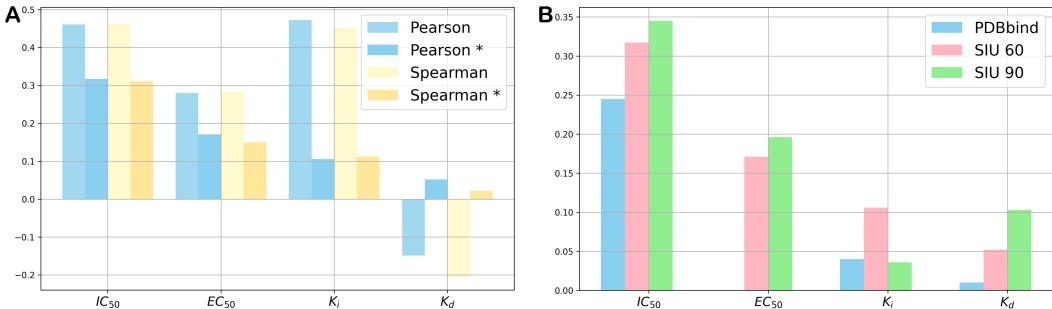

Figure 5: **(A)** Pearson and Spearman correlations for various label types, calculated both before and after grouping by PDB IDs. **(B)** Pearson correlations after grouping PDB IDs for different assay types trained on different datasets.

**may not effectively capture a model's ability to differentiate between molecules targeting the same protein.** Such discriminatory capacity is crucial in drug discovery, emphasizing the importance of focusing on molecular interactions specific to each target rather than general correlations across diverse targets. This underscores the necessity of our dataset, which measures correlation within the same PDB IDs, providing a more relevant assessment of a deep learning model's utility in drug discovery. Also, as the correlation is calculated within same pocket, **it cannot be overfitted with pocket only information**, as it will result in similar prediction results for different molecules, which would lead to NaN when calculating the pearson or spearman correlation.

**Effectivness of training on our dataset.** We compare models trained on the PDBbind 2020 dataset with those trained on SIU versions 0.6 and 0.9. Notably, the PDBbind 2020 dataset was used in its entirety, without implementing any filtering techniques to exclude pockets similar to those in the test set. As illustrated in Table 10 and Figure 5, models trained on the SIU datasets outperform those trained on PDBbind, despite the latter's lack of homology removal. **This underscores the effectiveness of our large-scale dataset in enhancing model learning for binding affinity prediction.** Also the 0.9 version gives a better performance compared to the 0.6 version, indicating the influence of removing homology and scaling law of the dataset. We also provide the results of using docked structures of PDBbind to train the model, which is shown in Appendix E.1. The results yield the same conclusion.

## 5 CONCLUSION

We identified and further analyzed the inherent biases present in mainstream bioactivity prediction tasks. These tasks tend to introduce bias in both the training and testing processes. During training, the bias arises due to the limited scale of small molecule-protein pairs; in most current datasets, there is only one small molecule associated with each protein pocket. This limits the model's ability to learn the underlying interactions between small molecules and protein targets, leading to overfitting to the value range of these protein pockets. Furthermore, during testing, existing metrics primarily assess the models' ability to discriminate between different protein pockets, neglecting their ability to rank various small molecules that interact with the same protein pocket. To address these critical challenges, we redefined the bioactivity prediction task by introducing a novel, large-scale, high-quality structural dataset with well-organized labels. We also developed new metrics that specifically evaluate a model's ability to rank different small molecules for each protein target. Our analysis, which included testing several classical models as baselines, demonstrates that our dataset can improve model performance. Moreover, our proposed metrics provide a more challenging and meaningful evaluation of bioactivity prediction models. Therefore, the task we introduced and the SIU dataset we created represent valuable contributions to the field.

ACKNOWLEDGMENTS

This work is supported by Beijing Academy of Artificial Intelligence (BAAI).

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

## APPENDIX

## LIST OF CONTENTS FOR APPENDIX

- A Detailed data usage description
- B General statistics and quality evaluation
- C Data construction methodology
- D Model training details
- E Additional experimental results
- F Other potential applications of the SIU dataset

## A    DETAILED DATA USAGE DESCRIPTION

### A.1    A PICKLE FILE THAT CONTAINS PROCESSED DATA AND LABELS FOR THE DATASET

The dataset is provided as a dictionary containing processed information. Each key represents a UniProt ID, and the corresponding value is a list of dictionaries. Each dictionary represents a data point and contains the following keys:

```
{
    source data : PDB ID and UniProt ID information,
    label       : A dictionary for different labels,
    including Ki, Kd, IC50, EC50,
    ik          : InChIKey of the ligand,
    smi         : SMILES of the ligand
}
```

### A.2    STRUCTURE FILES

The structure files are organized as follows:

```
DIR
 uniprotid
    pdb_id
       pocket_pdb_file
       inchikey1
          pose1.sdf
          pose2.sdf
          pose3.sdf
       inchikey2
          pose1.sdf
          pose2.sdf
          pose3.sdf
     ...
   ...
```

Users can use this structure to process their own data for their models.

### A.3    DATA SPLITS

To ensure both flexibility and robust evaluation, we provide a range of predefined dataset splitting strategies. Splits for the training and test sets of our task are provided. However, we do not impose a fixed split for the dataset, allowing users the flexibility to perform their own splits. These options provide users with the flexibility to evaluate models under various levels of sequence and structural

similarity constraints. A 10-fold cross-validation setting is also provided, as it provides a reliable and unbiased framework for machine learning model evaluation by ensuring diverse training and testing scenarios.

The updated version of the dataset includes five different predefined splitting strategies:

1. **90% Sequence Identity Filter:** A fixed test set is provided, and proteins with a sequence identity greater than 90% to the test set are removed from the training set.

2. **60% Sequence Identity Filter:** A fixed test set is provided, and proteins with a sequence identity greater than 60% to the test set are removed from the training set.

3. **60% Sequence Identity + Structural Similarity Filter:** A fixed test set is provided, and proteins with a sequence identity greater than 60% to the test set are removed from the training set. Additionally, protein pockets with a structural similarity greater than 20% to the test set are also excluded from the training set.

4. **30% Sequence Identity Filter:** A fixed test set is provided, and proteins with a sequence identity greater than 30% to the test set are removed from the training set.

5. **10-Fold Cross-Validation Split:** The dataset is divided into 10 clusters. Any pair of proteins with a sequence identity greater than 60% are placed within the same cluster. This split can be used for 10-fold cross-validation.

**Splitting methods.** To ensure the generalizability of the experimental findings with SIU, we employed a manual curation approach for dataset splitting. We selected a set of 10 representative protein targets to serve as the test set. These targets were intentionally chosen to cover a diverse range of protein classes, including well-known drug targets such as G-Protein Coupled Receptors (GPCRs) (Hauser et al., 2017), kinases (Attwood et al., 2021; Cohen et al., 2021), and cytochromes (Danielson, 2002). This selection strategy was designed to encompass the bioactivity landscape across various protein functionalities, thereby enhancing the applicability of our results to a wider range of potential drug discovery applications. We conducted non-homology analyses at two levels, 0.6 and 0.9, to ensure the independence and diversity of the training and test sets. For both versions 0.9 and 0.6, we have 21,528 data pairs allocated for testing. Specifically, version 0.9 includes 1,250,807 data pairs for training and validation, while version 0.6 includes 386,330 data pairs for these purposes.

Table 3: The curated test set of 10 protein targets, covering a diverse range of protein classes and displaying an even distribution of small molecule-pocket pair counts.

| UniProt | Gene name | Class | Small molecule-pocket pair count |
|---|---|---|---|
| P61073 | CXCR4_HUMAN | GPCR | 1376 |
| P42866 | OPRM_MOUSE | GPCR | 2379 |
| Q00535 | CDK5_HUMAN | Kinase | 2189 |
| Q04759 | KPCT_HUMAN | Kinase | 2320 |
| P15538 | C11B1_HUMAN | Cytochrome | 2427 |
| P13631 | RARG_HUMAN | Nuclear Receptor | 1888 |
| Q12879 | NMDE1_HUMAN | Ion Channel | 2144 |
| Q9UGN5 | PARP2_HUMAN | Epigenetic | 2251 |
| Q86WV6 | STING_HUMAN | Others | 2495 |
| Q96SW2 | CRBN_HUMAN | Others | 2059 |

**Test set construction.** To ensure the robustness and generalizability of the experimental findings with SIU, we meticulously curated a test set composed of 10 protein targets, as listed in Table 3. These targets were selected to represent a wide range of protein classes, including G-Protein Coupled Receptors (GPCRs), kinases, cytochrome, nuclear receptor, ion channel, epigenetic, and others, ensuring broad coverage of the bioactivity landscape. For example, "C11B1_HUMAN" belongs to the cytochrome P450 family, which is involved in the metabolism of various drugs (Bureik et al., 2002; Denisov et al., 2005). "RARG_HUMAN" belongs to the Nuclear Receptor family, with drugs like bexarotene used for certain cancers (Altucci et al., 2007; Qu & Tang, 2010). "NMDE1_HUMAN"

represents the NMDA receptor, a critical glutamate receptor in neurons implicated in various neuro-logical disorders, with memantine being an approved NMDA receptor antagonist for moderate to severe Alzheimer's disease (Mori & Mishina, 1995; Reisberg et al., 2003). Including these targets across various functionalities enhances the applicability of our results in drug discovery.

# B  GENERAL STATISTICS AND QUALITY EVALUATION

## B.1  GENERAL STATISTICS OF THE DATASET

SIU represents a large-scale, high-quality dataset of small molecule-protein interactions, meticulously organized to facilitate unbiased bioactivity prediction, both PDB-wise and assay-type-wise. The dataset comprises a total of 5,342,250 conformations. Each instance in the dataset provides detailed information about small molecule-protein interactions, including the coordinates and element types of each atom in the small molecule and the corresponding pockets of each interaction. Additionally, the assay value and type of each conformation, along with other critical information, are carefully obtained and retained from the original bioactivity databases. This includes the UniProt ID and PDB ID of the protein pockets, as well as the InChI keys (Heller et al., 2015) and SMILES Weininger (1988); Weininger et al. (1989) notations of the small molecules.

Table 4: The label count for 4 representative label types in SIU total, SIU 0.9, and 0.6 versions.

| | SIU 0.9 version | | | | SIU 0.6 version | | | |
|---|---|---|---|---|---|---|---|---|
| | Total | Train | Valid | Test | Total | Train | Valid | Test |
| $MTL$ | 1272335 | 1125727 | 125080 | 21528 | 407858 | 347697 | 38633 | 21528 |
| $IC_{50}$ | 962063 | 854230 | 94859 | 12974 | 320594 | 276969 | 30651 | 12974 |
| $EC_{50}$ | 97952 | 84067 | 9508 | 4377 | 32842 | 25675 | 2790 | 4377 |
| $K_i$ | 198091 | 175442 | 19447 | 3202 | 47946 | 40188 | 4556 | 3202 |
| $K_d$ | 54570 | 47347 | 5347 | 1876 | 17509 | 14003 | 1630 | 1876 |

## B.2  FURTHER EVALUATION OF LABEL VALUE DIFFERENCES ACROSS LABEL TYPES

To further support our rationale for dividing predictions by label type, we consider the potential coupling relationships between label types and proteins, which could lead to distributional differences across label types. To address this, we selected only pairs that contain all four label types, ensuring that the corresponding list of proteins is identical for each label type.

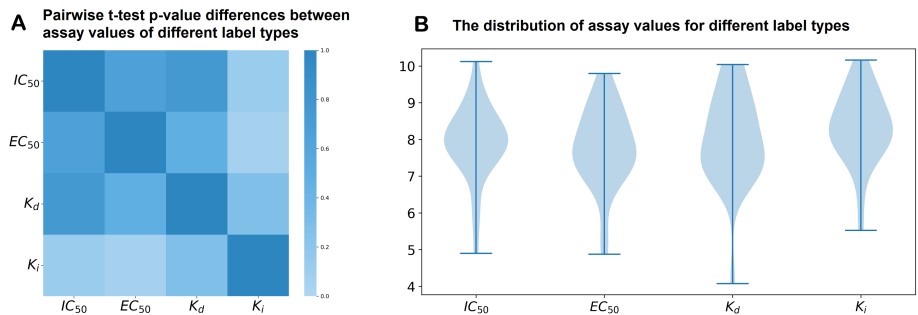

Figure 6: **Differences in assay values across different label types.** **(A)** Pairwise differences in mean assay values among these label types, as shown by a heatmap of pairwise t-test p-values. Smaller p-values (lighter colors) indicate significant differences. **(B)** Violin plots illustrate the distributions of label values for different label types.

Despite this refinement, the figures still reveal distributional differences across label types. Further-more, by ensuring that each pair includes all four label types, we calculated the Pearson correlation between label types for a given protein. The results are presented in Table 5.

Thus, we continue to believe that separating these types into distinct tasks is a worthwhile approach.

Table 5: Correlation matrix for $IC_{50}$, $EC_{50}$, $K_d$, and $K_i$ values. The table shows pairwise correlations between the different bioactivity measures.

| | $IC_{50}$ | $EC_{50}$ | $K_d$ | $K_i$ |
|---|---|---|---|---|
| $IC_{50}$ | 1 | 0.0773 | -0.0080 | 0.5849 |
| $EC_{50}$ | 0.0773 | 1 | 0.3752 | 0.2673 |
| $K_d$ | -0.0080 | 0.3752 | 1 | 0.0463 |
| $K_i$ | 0.5849 | 0.2673 | 0.0463 | 1 |

## B.3 FURTHER EVALUATION OF LABEL VALUE DIFFERENCES ACROSS PROTEIN POCKETS.

In Figure 4 we have vitualized the label value ranges difference in ten targets. To further provide a more comprehensive and convincing demonstration, in Figure 7 we provide here a heatmap and violin plot across more randomly sampled protein targets following the same logic. These representations further confirm that our findings remain consistent and valid in this broader context.

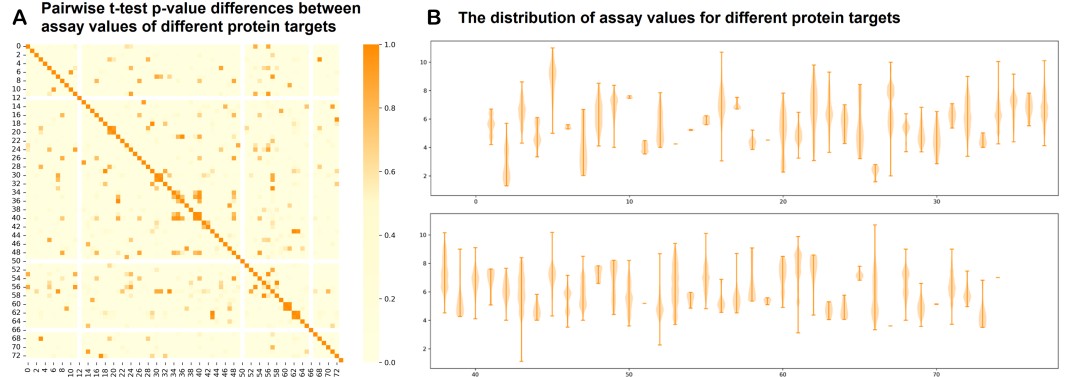

Figure 7: **Differences in assay values across more random-sampled protein targets. (A)** Pairwise differences in mean assay values among these protein targets, as shown by a heatmap of pairwise t-test p-values. Smaller p-values (lighter colors) indicate significant differences. **(B)** Violin plots illustrate the distributions of label values for different protein targets.

## B.4 FURTHER EVALUATION OF STRUCTURAL DATA QUALITY

We implemented rigorous quality control measures, such as the consensus method described earlier, to enhance the overall reliability of the generated structrual data. The intention of our work is not to propose a new docking algorithm but rather to create a dataset that facilitates redefining the bioactivity prediction task. In this effort, our focus was on selecting robust and widely accepted mainstream docking methods.

**Further evaluation in a cross-docking scenario.** We conducted an experiment focusing on one target with 20 PDB structures with co-crystal ligands within the same binding site. After aligning these complex structures, we performed cross-docking on them and calculated the RMSD between the docked ligand poses in all holo pockets and the corresponding co-crystal ligand poses.

As shown in Table 6, there is a significant improvement across various metrics for the docking poses that pass the filter compared to those that fail, highlighting the effectiveness of our consensus algorithm in ensuring the quality of docking poses.

Table 6: RMSD statistics for structures that passed or failed the filter. The table shows mean and median RMSD values, as well as the percentage of structures with RMSD below 1 Å , 2 Å, and 3 Å.

| Metric | RMSD Mean | RMSD Median | Percentage < 1 Å RMSD | Percentage < 2 Å RMSD | Percentage < 3 Å RMSD |
|---|---|---|---|---|---|
| Passed the Filter | 3.24 | 2.02 | 17.12% | 49.42% | 61.87% |
| Failed the Filter | 5.76 | 5.24 | 1.80% | 6.26% | 16.44% |

### B.5 FURTHER COMPARISON WITH PDBBIND

Additionally, as illustrated in Table 7, the dataset encompasses over 1,385,201 assay labels, each derived from corresponding wet-lab bioactivity experiments, ensuring the reliability and accuracy of the bioactivity information. SIU includes 1,720 diverse protein targets, with each protein potentially possessing multiple distinct binding pockets, verified through rigorous deduplication methods, resulting in a total of 9,662 unique pockets. The dataset also features a substantial and diverse collection of small molecules, totaling 214,686, across all pockets. Importantly, we have only included protein pocket-small molecule pairs confirmed to be active or inactive through wet-lab experiments, amounting to over 1,291,362 million pairs.

Table 7: Comparison of PDBbind and SIU datasets, showing the number of pocket-molecule pairs, average molecules per pocket, and unique pockets and molecules.

| Dataset | Pocket-molecule pairs | Avg. molecules per pocket | Unique pockets | Unique molecules |
|---|---|---|---|---|
| PDBbind | 19,443 | 1 | 19,443 | 19,443 |
| SIU | 1,312,827 | 137.6 | 9,544 | 214,686 |

### B.6 COMPARISON WITH FEP DATASET

By predicting binding free energy, FEP can be considered a bioactivity prediction method to some extent. However, handling one pocket at a time represents an inherent limitation of these methods, as they do not actively group data to address biases when evaluating the model. When evaluating models with FEP dataset, it is still necessary to calculate metrics as an average across different targets to assess general performance. While reporting metrics for individual targets can highlight performance of one model on specific targets, it does not provide a clear comparison of overall performance. Another limitation of the FEP dataset is the small number of targets and the limited number of small molecules per target. While this setup might serve as a held-out test for machine learning models, it addresses a different aspect from the motivation of this paper, which aims to provide a more comprehensive and unbiased evaluation.

## C DATA CONSTRUCTION METHODOLOGY

### C.1 DATA PREPROCESSING METHOD DETAILS

**Label data extraction and cleaning.** We retrieved non-structural bioactivity data from ChEMBL (Release 33) and BindingDB (version 202404), applying rigorous filtering to refine the dataset. For ChEMBL, data were extracted via SQLite, focusing on records with an assays.confidence_score of 9, targeting a single protein, and classified as binding (B) or functional (F). We included entries with non-null values for activities.standard_relation and activities.standard_value, where activities.standard_units were 'pM', 'nM', or 'μM'. Further refinement was applied to activities.activity_comment to capture specific biological activity descriptions, and molecular weight (compound_properties.mw_freebase) was restricted to between 150 and 650 Da. For small molecules, stringent filtering criteria were used to exclude non-drug-like entities. We selected molecules with a molecular weight of 150-650 Da, containing at least one carbon atom, and having a minimum of nine heavy atoms. Each small molecule retained its original IUPAC International Chemical Identifier (InChI) keys and Simplified Molecular Input Line Entry System (SMILES) notations. From BindingDB, we extracted relevant data from the .tsv file and standardized protein target information using UniProt IDs. The ChEMBL and BindingDB datasets were merged by matching InChI keys for small molecules with UniProt IDs for protein targets, ensuring accurate alignment of bioactivity labels with their corresponding small molecule-protein interactions.

**Label data deduplication.** To deduplicate small molecules, the Extended Connectivity Fingerprint (ECFP) (Rogers & Hahn, 2010) was utilized. ECFP is a type of molecular fingerprint commonly used in cheminformatics to represent the structural features of small molecules. It extends the circular fingerprint method by encoding a molecule into a fixed-length bit string that reflects its substructural characteristics. ECFPs are widely applied in various computational chemistry tasks,

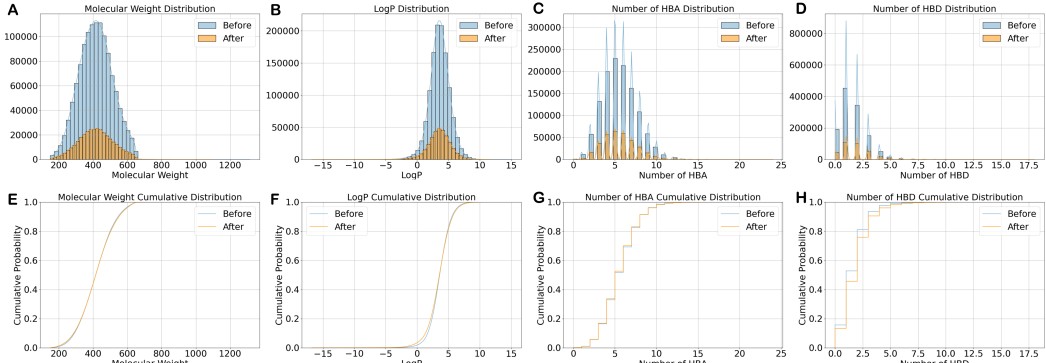

Figure 8: **Property Distributions of Small Molecules Before and After Deduplication.** Key characteristics such as molecular weight, water-oil partition coefficient (logP), hydrogen bond acceptor (HBA) count, and hydrogen bond donor (HBD) count remain consistent, despite a reduction in the total number of small molecules. **(A-D)** Distributions of small molecules across different ranges of molecular properties. The y-axes represent the number of small molecules within specific property ranges. **(E-H)** Cumulative distributions of the same molecular properties, with the y-axes indicating cumulative probabilities.

such as virtual screening, similarity searching, and machine learning. In this work, small molecules with high structural similarity (Tanimoto similarity $> 0.8$) were grouped into clusters. One or more representatives from each cluster, prioritized based on bioactivity, were selected to deduplicate the dataset while maintaining structural diversity. Notably, only small molecules within pockets where the molecule count exceeds the 90th percentile threshold (2,146) across all pockets were subjected to deduplication. As illustrated in Figure 8, the diversity of small molecules was preserved following the application of this deduplication method. Fast Local Alignment of Protein Pockets (FLAPP) (Sankar et al., 2022) is a program that used to calculate the structure similarity between two protein pockets. We remove structurally redundant pockets using this program. The threshold is set to 0.9.

**Pocket definition.** Biologically speaking, a protein typically exhibits a limited number of distinct binding sites, defined as regions accommodating specific ligands. It's important to differentiate between these binding sites and the "pockets" we identify, which are characterized by the ligands used for their definition rather than their spatial distribution on the protein surface. In essence, multiple pockets can potentially reside within a single binding site, since a lot of binding sites have complex structures with multiple ligands therefore have a lot of PDB IDs. Different PDB IDs represent distinct resolved structures for a single protein, all associated with the same UniProt ID. **For each PDB ID, a single pocket was extracted, defined as the region centered on the co-crystal ligand within a 15 Å radius.**

**Mapping small molecule to pocket.** While our method can be considered a form of cross-docking, it fundamentally differs from the well-known CrossDocked2020 dataset (Francoeur et al., 2020). Specifically, we use only small molecules with experimentally determined bioactivity values to dock to their corresponding protein targets. This ensures that each small molecule is docked to at least the correct protein target. Additionally, our consensus docking not only generates structural data but also serves as a reverse strategy for identifying the appropriate binding pocket, thereby enhancing the reliability of the pocket assignments.

## C.2 DOCKING METHOD DETAILS

Constructing a structural dataset of this scale required careful optimization of docking parameters to balance performance and computational resource efficiency. Ligands from bioactivity databases, provided as 1D representations (SMILES), were first processed with Glide LigPrep to generate up to 32 stereoisomers per molecule, accounting for potential variations in stereochemistry at chiral centers. Ionization states were predicted at physiological pH to ensure accurate representation of charged forms. Multiple conformations were prepared for each small molecule to account for molecular

flexibility. Protein structures were preprocessed by predicting their ionization states, removing water molecules, and preparing docking grid files centered on the co-crystal ligand. Docking was performed with default settings. The resulting conformations were filtered through a consensus pipeline to select the final set of conformations included in the SIU dataset.

## D   MODEL TRAINING DETAILS

For GNN Model, we use the same model in Atom3D (Townshend et al., 2021). We train the model using one NVIDIA A100 GPU. The batch size is 256, the max number of epochs is 20, the optimizer is Adam, the learning rate is 1e-3.

For 3D-CNN Model, we use the same model in Atom3D. We train the model using one NVIDIA A100 GPU. The batch size is 256, the max number of epochs is 20, the optimizer is Adam, the learning rate is 1e-4.

For Uni-Mol model, we use the pretrained model weights provided. The pretrained molecular encoder and pocket encoder outputs are concatenated and passed through a four-layer Multi-Layer Perceptron (MLP) with hidden dimension 1024, 521, 256, 128. We use four NVIDIA A100 GPU to train the model. The batch size is 384, the max number of epochs is 50, the optimizer is Adam, the learning rate is 1e-4.

For ProFSA model, we use the pretrained model weights provided. The pretrained molecular encoder and pocket encoder outputs are concatenated and passed through a four-layer MLP with hidden dimension 1024, 521, 256, 128. We use four NVIDIA A100 GPU to train the model. The batch size is 384, the max number of epochs is 50, the optimizer is Adam, the learning rate is 1e-4.

## E   ADDITIONAL EXPERIMENTAL RESULTS

### E.1   ADDITIONAL RESULTS FOR USING THE DOCKED STRUCTURE OF PDBBIND

We conduct additional results using the docked structures instead of the original complex of PDBbind. The result is shown in Table 8.

Table 8: Results for single task training with different label types. We show the results with Uni-Mol model on PDBbind dataset, our SIU 0.6 version and 0.9 version dataset. Pdocked is the model trained on PDBbind but with the docked structure instead of original complex structure.

|  | Train Set | RMSE | MAE | Pearson | Pearson* | Spearman | Spearman* |
|---|---|---|---|---|---|---|---|
| $IC_{50}$ | PDBbind | 1.575 | 1.279 | 0.430 | 0.245 | 0.425 | 0.229 |
|  | Pdocked | 1.600 | 1.304 | 0.439 | 0.241 | 0.436 | 0.229 |
|  | SIU 0.6 | 1.407 | 1.138 | 0.461 | 0.317 | 0.463 | 0.311 |
|  | SIU 0.9 | 1.357 | 1.099 | 0.470 | 0.345 | 0.474 | 0.347 |
| $EC_{50}$ | SIU 0.6 | 1.400 | 1.163 | 0.280 | 0.171 | 0.284 | 0.150 |
|  | SIU 0.9 | 1.340 | 1.096 | 0.384 | 0.196 | 0.379 | 0.142 |
| $K_i$ | PDBbind | 1.315 | 1.085 | 0.368 | 0.040 | 0.323 | 0.026 |
|  | Pdocked | 1.301 | 1.059 | 0.365 | 0.035 | 0.311 | -0.001 |
|  | SIU 0.6 | 1.255 | 1.034 | 0.407 | 0.106 | 0.452 | 0.112 |
|  | SIU 0.9 | 1.235 | 1.017 | 0.385 | 0.036 | 0.452 | 0.041 |
| $K_d$ | PDBbind | 1.565 | 1.308 | 0.041 | 0.010 | 0.004 | 0.006 |
|  | Pdocked | 1.447 | 1.231 | -0.015 | 0.034 | -0.062 | 0.011 |
|  | SIU 0.6 | 1.389 | 1.192 | -0.049 | 0.052 | -0.206 | 0.022 |
|  | SIU 0.9 | 1.364 | 1.141 | 0.033 | 0.103 | -0.082 | 0.065 |

### E.2   ADDITIONAL RESULTS FOR DIFFERENT SPLITTING

The results for combining 60% sequence identity and 0.2 FLAPP structure similarity are shown in Table 9.

The results for 10 folds split are shown in Table 10.

Those results pertain to the Uni-Mol model.

Table 9: Results for single task training with different label types. We show the results with Uni-Mol model on SIU that is based on our original 0.6 split, with FLAPP to remove additional similar pockets from the train set.

|  | RMSE | MAE | Pearson | Pearson* | Spearman | Spearman* |
|---|---|---|---|---|---|---|
| $IC_{50}$ | 1.389 | 1.117 | 0.411 | 0.353 | 0.412 | 0.355 |
| $EC_{50}$ | 1.399 | 1.164 | 0.212 | 0.165 | 0.232 | 0.154 |
| $K_i$ | 1.318 | 1.096 | 0.456 | 0.122 | 0.443 | 0.118 |
| $K_d$ | 1.349 | 1.154 | -0.087 | -0.042 | -0.123 | -0.14 |

Table 10: Results for single task training with different label types. We show the results with Uni-Mol model on SIU that is splitter in to 10 folds based on sequence identity. The result is trained on 8 folds, validate on 1 fold, and tested on another fold.

|  | RMSE | MAE | Pearson | Pearson* | Spearman | Spearman* |
|---|---|---|---|---|---|---|
| $IC_{50}$ | 1.332 | 1.046 | 0.421 | 0.216 | 0.408 | 0.203 |
| $EC_{50}$ | 1.322 | 1.123 | 0.494 | 0.227 | 0.520 | 0.228 |
| $K_i$ | 1.545 | 1.281 | 0.356 | 0.143 | 0.355 | 0.133 |
| $K_d$ | 1.678 | 1.369 | 0.474 | 0.149 | 0.347 | 0.153 |

# F  OTHER POTENTIAL APPLICATIONS OF THE SIU DATASET

The SIU dataset offers a wealth of opportunities for advancing drug discovery by addressing a wide range of applications. Beyond its core use in unbiased bioactivity prediction, the dataset is being extended to support pairwise ranking tasks, ensuring that comparisons are restricted to docking poses from the same PDB structure and bioactivity data with consistent label types. Additionally, SIU can be leveraged as a virtual screening resource. Another promising application lies in training models for molecular optimization based on pairwise data, distinguishing molecules with high and low bioactivity. Furthermore, SIU serves as a supplementary dataset for docking pose prediction tasks, providing a reliable resource. Finally, the automated pipeline demonstrated in this work showcases the feasibility and scalability of generating large-scale, high-quality datasets. Future efforts could refine these methods to construct even larger datasets, enhancing the understanding of small molecule-protein interactions and accelerating progress in AIDD.

