# OpenReview forum: "Redefining the task of Bioactivity Prediction"
_ICLR.cc/2025/Conference — ICLR 2025 Poster_

### Official Review · Reviewer_6s1k · 2024-10-17

**Soundness:** 4
**Presentation:** 3
**Contribution:** 3
**Rating:** 8
**Confidence:** 4

**Summary:**

The authors present a new dataset for bioactivity prediction together with a more fair metric for evaluation of ML models for bioactivity prediction. The authors present valuable insights regarding the weaknesses of current state of the art which convincingly motivates their work. The new carefully curated dataset together with the new metric tackles the identified weaknesses of current SOTA.

**Strengths:**

1. Identification of the issues with current SOTA: overfitting to protein pockets due to small diversity of current datasets in terms of containing different small molecules docked to a single protein pocket; issues in reporting the overall correlation instead of computing the partial correlations per target and pooling the partial correlations which can help hide the problem of overfitting to protein pockets

2. Introduction of a new large dataset for bioactivity prediction

3. Careful curation of the dataset using a custom method of majority vore over different docking tools used for generation of the docked pose. Separation by different essay types is also useful.

**Weaknesses:**

1. There are few smaller unclarities in the text:
- On line 174, I did not understand what it is meant by "anti-logged"
- I did not find what method was used for generation of the small molecule conformers, it would be good to clarify this near the statement "Initial 3D conformations for the small molecules were generated prior to docking."
- Figure 4 is discussed in the text, but the caption could use more detailed description. Just by looking at the figure it is not clear what we are looking at.
- The statement “We conducted non-homology analyses at two levels, 0.6 and 0.9,” is not clear. Did the authors check for sequence identity less than 60/90 pct. between test and train set? Or the complement of 40/10 pct.? Please clarify.

2. No mention of the Papyrus dataset ( https://jcheminf.biomedcentral.com/articles/10.1186/s13321-022-00672-x ). It should be discussed in related work and authors should compare SIU to Papyrus in all relevant aspects, such as size, data origin and level of curation.

3. Detailed, practical description of the dataset, which would facilitate its usage by developers of ML models is missing. I suggest putting to supplementary material more detailed and practical description of what is provided in the dataset. PDB files containing the structure of the protein-ligand complexes as docked by some of the docking software? And the labels are just a single scalar value per datapoint? Please explain in detail.

**Questions:**

1. The presented majority vote system over the 3 docking system is interesting even from the point of view of docking. What percentage of poses passed this filter? Did you notice some 2 of the 3 tools to tend to agree more with each other? It would be interesting to benchmark this method against the individual tools on some docking dataset (e.g. PDBBind) - for example take the performance of the 3 individual tools evaluated on PDBBind and compare it to the performance of the best of the 3 tools evaluated just over the datapoints which passed the majority vote filter. This benchmarking is not a strict requirement for the rebuttal period from me, but I would be really interested to see it.

2. Could you please describe in more detail how were the models trained for the bioactivity prediction? E.g. the docking model UniMol is described in SuppMat to have an MLP on top of it. What is the input and what is the output of such model. Is the MLP trained to predict a single scalar value for a protein-ligand pair? Please explain.

---

> ### Author Response · Authors · 2024-11-20
>
> Thank you for your valuable review. We greatly appreciate your recognition of our contribution in identifying issues with the existing task and our efforts in carefully constructing a reliable dataset for the task. We aim to address any unclarities in our paper and respond to your questions thoroughly.
>
>
> ## Regarding unclarities in the text
>
> Thanks for bringing up the unclarities in the text, we've checked all of these and updated with the correct or more detailed terms. We meant "converted to their negative logarithm" by "anti-logged". For initial 3D conformation generation of small molecules we use the Glide LigPrep module with default settings. For Figure 4, we've updated the figure itself to provide the general information for each panel of the figure and changed the caption to "Figure 4: Differences in assay values across label types and protein targets. (A) The mean assay values vary among representative label types, as shown by a heatmap of pairwise t-test p-values. Smaller p-values (lighter colors) indicate significant differences. (B) Violin plots illustrate the distributions of label values for different label types. (C) Differences in mean assay values among ten protein targets. (D) The distributions of label values for different protein targets." The updated figure can also be found at https://anonymous.4open.science/r/SIU_ICLR-C475/distributions_update.png
>
> ## Regarding data split
>
>
> We apologize for any lack of clarity in our description. For the 0.6 and 0.9 non-homology levels, we set sequence identity thresholds at 60% and 90%, respectively. Specifically, the 0.6 non-homology level ensures that no sequences in the training and test sets share more than 60% sequence identity. Sequence identity is calculated based on alignment scores generated using the pairwise2.align function in BioPython.
>
> ## Regarding papyrus dataset
>
> Thank you for bringing this paper to our attention. Papyrus is indeed an excellent dataset that also aims to provide a high-quality resource for bioactivity prediction tasks. We have incorporated the suggested discussion into the "Related Work" section.
>
> ## Degarding description to facilitate dataset usage
>
> Thank you for pointing out that a more detailed description to facilitate dataset usage is needed. We have added it to the appendix and here is the content:
>
> ### split files
>
> Splits for the training and test sets of our task are provided. However, we do not impose a fixed split for the dataset, allowing users the flexibility to perform their own splits. The updated version of the dataset includes four different predefined splitting strategies:
>
> 1. 90% Sequence Identity Filter: A fixed test set is provided, and proteins with a sequence identity greater than 90% to the test set are removed from the training set.
> 2. 60% Sequence Identity Filter: A fixed test set is provided, and proteins with a sequence identity greater than 60% to the test set are removed from the training set.
> 3. 60% Sequence Identity + Structural Similarity Filter: A fixed test set is provided, and proteins with a sequence identity greater than 60% to the test set are removed from the training set. Additionally, protein pockets with a structural similarity greater than 20% to the test set are also excluded from the training set.
> 4. 10-Fold Cross-Validation Split: The dataset is divided into 10 clusters. Any pair of proteins with a sequence identity greater than 60% are placed within the same cluster. This split can be used for 10-fold cross-validation.
>
> These options provide users with the flexibility to evaluate models under various levels of sequence and structural similarity constraints.
>
> ### A pickle file that contains processed data and label for the dataset
>
> It is a dictionary that contains the processed information of the dataset. Each key is a UniProt ID, and corresponding value is a list of dictionaries. Each dictionary is a data point and has following keys:
>
> {
>
> source data : PDB ID and UniProt ID information
>
> label : a dictionary for different labels, including Ki, kd, ic50, ec50. Each label is a scalar value.
>
> ik : InChI key of molecule
>
> smi : SMILES of molecule
>
> }
>
> ### structure files
>
> It is a dictionary of the following format:
>
> DIR
>
> ├── uniprot id
>
> │   ├── pdb id
>
> │   │   ├── pocket pdb file
>
> │   │   ├── inchikey1
>
> │   │   │   ├── pose1 sdf file
>
> │   │   │   ├── pose2 sdf file
>
> │   │   │   ├── pose3 sdf file
>
> │   │   ├── inchikey2
>
> │   │   │   ├── pose1 sdf file
>
> │   │   │   ├── pose2 sdf file
>
> │   │   │   ├── pose3 sdf file
>
> │   │   ...
>
> │   ...
>
> Users can use this to process their own data for their own models.

---

> ### Author Response · Authors · 2024-11-20
>
> ## Regarding filter passing percentage
>
> Thank you for considering our method interesting. We can provide the percentage of poses that passed our consensus filter, calculated from a random 10% subset of our data. For the pose generated by Glide, the pass rate is 36%.
>
> We apologize for not being able to provide the actual number, as we have deleted some parts of the original output from the docking software programs to reduce storage costs. Additionally, we provide further analysis on the PDBbind dataset in the following section.
>
> ## Regarding docking benchmark
>
> Thank you for pointing out that we should benchmark our proposed voting mechanism on a docking dataset to evaluate its efficiency. We conducted this evaluation on the PDB refined dataset, which contains 5318 protein-small molecule complex structures. We compared the three docking softwares, and we calculated the average RMSD for poses that passed the voting filter and those that did not for each software. The results are shown in Table 1:
>
> ### Table 1: Comparison of Docking Software Using Voting Mechanism
>
> | Method | Avg RMSD (Passed Filter) | Avg RMSD (Failed Filter) | Percentage of Molecules Passing Filter |
> |--------|--------------------------|--------------------------|-----------------------------------------|
> | Glide  | 2.04                     | 5.08                    | 82.70%                                  |
> | Gold   | 2.23                     | 6.00                    | 81.52%                                  |
> | Vina   | 3.07                     | 5.81                    | 56.16%                                  |
>
>
> From these results, it is evident that for each docking software, the molecules passing the filter exhibit a significantly lower average RMSD to the true co-crystal pose compared to those that do not. This demonstrates the effectiveness of our method in ensuring the quality of the final poses.
>
> We also used the same dataset to analyze which two docking softwares tend to agree with each other more. The results are presented in Table 2:
>
>
> ### Table 2: Agreement Between Docking Software
>
> |         | Glide   | GOLD    | Vina    |
> |---------|---------|---------|---------|
> | Glide   | -       | 23.2%   | 18.3%   |
> | Gold    | 23.2%   | -       | 18.1%   |
> | Vina    | 18.3%   | 18.1%   | -       |
>
>
> ## Regarding the models
>
> The MLP is trained to predict a single scalar value for a protein-ligand pair. The input of such MLP is the concatenation of the CLS pocket embedding from the pocket encoder, and the CLS molecule embedding from the molecule encoder. We have added it to the paper.

---

> ### Comment · Reviewer_6s1k · 2024-11-20
>
> Thank you very much for addressing most of the concerns.
>
> I still have two points:
>
> 1) What I meant by the docking experiment is to evaluate and report the performance of Glide, Gold and Vina on the PDBBind dataset (what is the percentage of predicted poses within 2A RMSD of the ground truth, averaged over PDBBind) and then also reporting this performance only over the datapoints where at least 2 of the softwares agreed. But this is not crucial for your paper, it just felt potentially interesting for the docking community, so decide freely whether to address this.
>
> 2) To me the sequence similarity thresholds of 0.6 and 0.9 seem to high. Could you please explain your choice? You can definitely have structural homologues with less than 0.6 sequence similarity. In my experience 0.3 or 0.4 are more common thresholds (e.g. AF2 uses them). Introducing a benchmark which would tolerate such leakage might slow down the progress on actually useful methods.

---

> > ### Author Response · Authors · 2024-11-21
> >
> > Thank you for your response. We are pleased to know that we have addressed most of your concerns.
> >
> > ## 1. Regarding the docking experiment
> >
> > Here is our experimental setup based on our understanding of your requirements:
> >
> > For each docking software, we generate several poses. For each generated pose, we check whether it achieves consensus with at least one pose generated by another software. If a pose meets this criterion, it is classified as a “passed pose”; otherwise, it is classified as a “failed pose.” We then calculate the percentage of poses with an RMSD less than 1 Å, 2 Å, and 3 Å compared to the ground truth pose. Finally, we compare the performance across three groups: passed poses, failed poses, and all poses combined. These is the results:
> >
> > | Method | Group           | Percentage RMSD < 1 | Percentage RMSD < 2 | Percentage RMSD < 3 |
> > |--------|-----------------|---------------------|---------------------|---------------------|
> > | GLIDE  | Passed Poses    | 40.19%             | 68.88%             | 79.66%             |
> > |        | Failed Poses    | 5.16%              | 12.54%             | 24.93%             |
> > |        | ALL             | 30.04%             | 52.55%             | 63.79%             |
> > | GOLD   | Passed Poses    | 40.31%             | 64.31%             | 75.71%             |
> > |        | Failed Poses    | 1.05%              | 4.13%              | 10.24%             |
> > |        | ALL             | 28.49%             | 46.19%             | 56.00%             |
> > | Vina   | Passed Poses    | 24.64%             | 45.87%             | 60.83%             |
> > |        | Failed Poses    | 1.10%              | 3.87%              | 13.60%             |
> > |        | ALL             | 12.04%             | 23.38%             | 35.54%             |
> >
> >
> > The results demonstrate that our consensus filter improves the percentage of successful docking poses across all software employed. Notably, while Glide is the most accurate method before applying the filter, its performance is also further enhanced with the “help” of other methods, highlighting the effectiveness of the consensus approach.

---

> ### Author Response · Authors · 2024-11-21
>
> ## 2. Regarding Sequence Identity Threshold
>
>
> That is a very good question. There are several reasons for us to choose such a threshold.
>
> From the experimental results, we observe that our task is not highly sensitive to the dataset split. Unlike previous bioactivity prediction tasks, which are sensitive to split levels due to model overfitting to the protein, our task effectively mitigates this issue. As shown in our analysis, models in earlier tasks could achieve high performance by overfitting to protein-only features, even though the task inherently requires both protein and molecule information. In contrast, our task emphasizes the relationship between a protein and multiple molecules, making it more difficult for models to overfit to the protein alone. Furthermore, the performance difference between the 0.9 sequence identity threshold and the 0.6 threshold is minor, which is why we have not further reduced the sequence identity threshold.
>
> Additionally, the results highlight the challenging nature of our task. The grouped Pearson correlation of the models is only around 0.2 to 0.3, or even around zero, compared to values as high as 0.8 in previous tasks like Atom3D LBA  for the same models. This demonstrates that data leakage or overfitting is not a major concern in our current split. Instead, the critical challenge for the bioactivity prediction community is to develop effective models capable of accurately predicting bioactivity and ranking small molecules correctly. Given this, retaining more data in the training set is more beneficial, as it allows models to learn from a larger dataset with more pocket-molecule pairs available.
>
> Furthermore, during the preprocessing step, we have already applied structural similarity filtering using Fast Local Alignment of Protein Pockets (FLAPP) [1], a program that used to calculate the structure similarity between two protein pockets, on the entire dataset to remove structurally similar pockets. The distribution of structure similarity scores between the train and test sets is shown here: https://anonymous.4open.science/r/SIU_ICLR-ADF1/flapp_scores.png. This distribution demonstrates that most pockets have a structural similarity of less than 0.2, indicating a clear distinction between train and test sets.
>
> Our primary goal is not to establish a benchmark but to introduce a new task with a novel dataset and evaluation methods that address the issues in previous tasks. For researchers aiming to further benchmark the extreme generalization ability of models, particularly if they believe overfitting has become an issue for our task, we provide a stricter version of the dataset split. In this stricter version, pockets from the training set with a structural similarity greater than 0.2 to any pocket in the test set are excluded. This is a highly stringent criterion and has been documented in the dataset description. A basic experiment results based on this split is also provided in Appendix G table 7.
>
>
> [1] Sankar, Santhosh, Naren Chandran Sakthivel, and Nagasuma Chandra. "Fast local alignment of protein pockets (FLAPP): a system-compiled program for large-scale binding site alignment." Journal of Chemical Information and Modeling 62.19 (2022): 4810-4819.

---

> > ### Comment · Reviewer_6s1k · 2024-11-21
> >
> > Thank you very much for the prompt response. The docking experiment is precisely what I was interested in.
> >
> > I still dont see why not to have a 0.3 similarity split as well, even if your tasks has some specifics which make it less prone to leakage.
> >
> > Nevertheless, I recommend an accept, I raise my score to 8.

---

> > > ### Author Response · Authors · 2024-11-22
> > >
> > > Thank you very much for your thorough review and for recognizing the value of our work. We plan to also include a split version with 30% similarity split when the dataset is released upon the acceptance of our paper.

---

### Official Review · Reviewer_MuEF · 2024-11-01

**Soundness:** 3
**Presentation:** 3
**Contribution:** 3
**Rating:** 6
**Confidence:** 4

**Summary:**

The paper proposes a new benchmark for training and testing of bioactivity prediction models, a critical task in computational biology. The curation of the training data involves filtering a set of more than one million bioactivities, cluster them into predictions around specific pockets and predict approximate structures for each of them. The curation of the test set the selection of a small subset of test complexes and the definition of the task as correlations within each target. Finally, the authors train a number of models on this new benchmark and highlight how the poor results on this assessment suggests more work needs to be done.

**Strengths:**

I believe that the curation of a dataset for protein small molecule activity prediction is very important. Current datasets are too small and used in the incorrect way as the authors also represent. In this work, the authors do a very extensive and carefully thought out job towards building such dataset.

**Weaknesses:**

Building and proposing a new benchmark constitutes a significant responsibility: once the benchmark is released and, as I hope, the community adopts it, it will become very hard to correct any wrong choice made at this stage and these may lead to a lot of work being largely wasted (as I believe it is happening with current benchmarks for this task). Therefore, my bar is not just having an improvement over what is public but making sure that researchers in the field would not have concerns about adopting such a dataset.

Based on this premise below I’m presenting a list of concerns that I would appreciate the authors responding to or addressing:

1. The authors only retained datapoints for which the docking methods were consistent in their majority. While this appears sensible, it also creates a bias in the kind of molecules that the model considers (intuitively only “easier” complexes). It could be interesting to ablate the addition of non-consensus poses in the training set or in the test set for some of the baselines.

2. The success proportion presented appears somewhat misleading. It is well known that redocking or holo-structure docking (that the authors use to validate the structure generation) is significantly easier than cross-docking (docking to the protein structure resolved when binding to a different protein, which the authors have to use when generating new structures). Therefore the experiments on the success should be performed on the cross docking task. This can be performed by taking from PDB structures of ligands bound to the same pocket.

3. Saying that structures generated with the docking programs are of high quality seems a bit exaggerated or at least lacking experimental backing (see comment above on the difference between redocking and cross-docking). Multiple works in the molecular docking community have shown significant issues with the apo or cross docked performance of the methods applied for docking in this work.

4. Assay types might be a bit misleading as the name of the “label type” as there are many different types of assays producing for example Kd values.

5. The definition of different pockets from PDB IDs of co-crystal ligands and the subsequent deduplication are very unclear. While this seems a relatively non-important detail it becomes critical once the authors decide to use this assignment to compute correlation over.

6. For protein ligand complexes from the bioactivity assays the way that these are assigned to a specific PDB ID / pocket is not clear to me. The available information is the Uniprot but then how does one go from having a specific Uniprot to deciding which of the PDB IDs corresponding to the different pockets to assign a complex to.

7. Basing the benchmark of the output of the docking models e.g. by filtering out complexes with non-consensus poses and grouping by predicted pocket, will not allow fair comparisons for non-structure based methods or methods based on a different docking algorithm. I would recommend the authors remove such filtering steps for the test set to avoid the applicability of the benchmark being limited.

8. Line 326 the authors say that the dataset is unbiased. Could they specify clearly what they mean by this term? Once again the claim does not seem fully justified.

9. In paragraph 3.3, the authors discuss the value of dividing the prediction for different assay types. To motivate this they apply statistical significance test to the mean of the distribution of different assay types. However, it is unclear to me why this would be a good motivation for that. Differences in averages might just mean that these assays were applied to different types of proteins, while what the authors would need to show to motivate this point is that if they were applied to the same complex the results would be different depending on the assay. This work was actually performed by a recent paper [1] which however showed no very clear outcomes when it comes to the differentiation between different assay types. An alternative way to try to demonstrate this point would be to put all datapoints in the same “task” and then train and evaluate a single task model.

10. I believe that saying that “In the traditional approach” or “conventional approach” people would look for pearson across different targets is misleading. This has been indeed unfortunately done recently in a number of ML papers, however, more established benchmarks in the field such as FEP+ do look at correlations computed per target.

I would raise my score if a convincing response or revision were made on these points.

[1] Combining IC50 or Ki Values from Different Sources Is a Source of Significant Noise. Landrum, Riniker.

**Questions:**

1. Figure 3: how is success ratio defined?

2. What type of assay does the test set contain? I assume it does not contain measurements across all different assay types. In this case how are spearman and person correlations computed for al assay types in the test set?

3. How will the dataset be released? What format and what license?

4. Is Pearson computed also on inactives? What proportion of the test and training set are inactive?

5. Do I understand correctly that the test set only has 10 targets?

6. In line 365 I assume the assay type is also used for the grouping, could the authors confirm?

---

> ### Author Response · Authors · 2024-11-20
>
> # In response to "weakness"
>
> We sincerely thank you for your thoughtful and constructive review. Your acknowledgment of its potential to be widely adopted by the community and its value in redefining this critical task is deeply encouraging. Our motivation has always been to contribute meaningfully to the community by addressing existing limitations and improving the definition of this task.
>
> We appreciate the concerns raised regarding the responsibility of this work. We have taking these concerns raised by the reviewer very seriously. we recognized that certain aspects of our methodology and rationale might not have been fully articulated. To address this, we are committed to providing detailed explanations with additional clarity in the updated version of our paper to alleviate any concerns.
>
> We also value your feedback on areas where improvements can be made and will incorporate these refinements to further strengthen our work. Your insights help us ensure that this work serves as a reliable and impactful resource for the community, and we are grateful for the opportunity to make it even better.
>
> Thank you once again for your support and for highlighting the potential contributions of this work.
>
> ## Regarding retaining only consensus filtered data points
>
> Thank you for your valuable comment regarding the potential bias introduced by retaining only data points with consistent docking results. We appreciate the opportunity to clarify our approach and its rationale.
>
> (1) The primary goal of our pipeline is to ensure the **quality of the conformations** used in our dataset. By focusing on cases where docking methods show consensus, we aim to reduce the bias introduced by any single docking software and improve the general quality of our structural data (this is further discussed in the following of our reply), which is crucial for **reliable bioactivity prediction tasks**.
>
> (2) While we acknowledge the concern about a potential bias toward "easier" complexes, it is important to note that the difficulty of docking is not directly correlated with the difficulty of the bioactivity prediction task, especially when framed as a machine learning problem. Docking failure or lack of consensus may arise from various factors, including limitations in docking algorithms, which do not necessarily reflect the inherent complexity of the bioactivity prediction task whose goal is to predict the corresponding bioactivity label.
>
> (3) The data points in our dataset are derived from well-established databases like ChEMBL and BindingDB. However, these databases inherently reflect research trends and human reporting biases, rather than the true distribution of bioactivity labels in nature. Therefore, our SIU dataset, like any derived from these sources, could not represent the real-world distribution.
>
> ## Regarding Cross docked performance
>
> We conducted an experiment in a cross-docking scenario, focusing on one target with 20 PDB structures with co-crystal ligands within the same binding site. After aligning these complex structures, we performed cross-docking on them and calculated the RMSD between the docked ligand poses in all holo pockets and the corresponding co-crystal ligand poses. The results are as follows:
>
>
> | Metric                   | Mean RMSD | Median RMSD | Percentage < 1 RMSD | Percentage < 2 RMSD | Percentage < 3 RMSD |
> |--------------------------|-----------|-------------|----------------------|---------------------|---------------------|
> | Passed the Filter        | 3.24      | 2.02        | 17.12%              | 49.42%             | 61.87%             |
> | Failed the Filter        | 5.76      | 5.24        | 1.80%               | 6.26%              | 16.44%             |
>
>
> As shown in the table, there is a significant improvement across various metrics for the docking poses that pass the filter compared to those that fail, highlighting the effectiveness of our consensus algorithm in ensuring the quality of docking poses.
>
> ## Regarding the docking quality
>
> Thank you for pointing this out. We recognize that our intended message was to emphasize that our pipeline achieves better quality compared to individual docking methods, as discussed earlier.
> We acknowledge the challenges faced by the docking community, but we would like to provide a few clarifications:
>
> (1) We did not use "apo" docking in our work.
>
> (2) The intention of our work is not to propose a new docking algorithm but rather to create a dataset that facilitates redefining the bioactivity prediction task. In this effort, our focus was on selecting robust and widely accepted mainstream docking methods to ensure acceptance by potential users.
>
> (3) We implemented rigorous quality control measures, such as the consensus method described earlier, to enhance the overall reliability of the generated dataset.

---

> > ### Author Response · Authors · 2024-11-20
> >
> > ## Regarding the assay type
> >
> > Thank you for pointing this out and suggesting a more suitable term. We appreciate your attention to detail, as it helps us present our work more clearly and accurately. We apologize for any confusion caused by the use of "assay type" in our previous writing. Based on your feedback, we have revised the manuscript and replaced all instances of "assay type" with "label type" to ensure better clarity and precision. Your thoughtful review is invaluable, and we are grateful for your contribution to improving our work.
> >
> > ## Regarding the detailing methods
> >
> > Thank you for highlighting this important point. While we included some relevant details on pocket definition and deduplication in the Appendix, your insightful feedback made us realize the critical importance of clarifying this aspect. In response, we have moved the information to the appropriate method section and expanded it with more detailed and precise descriptions to ensure greater clarity.
> >
> > ## Regarding the mapping to pocket
> >
> > Thank you for pointing this out. Indeed, such mapping information is not easy to obtain. While our method can be considered a form of cross-docking, it fundamentally differs from the well-known CrossDocked2020 dataset [1]. Specifically, we use only small molecules with experimentally determined bioactivity values to dock to their corresponding protein targets. This ensures that each small molecule is docked to at least the correct protein target. Additionally, our consensus docking not only generates structural data but also serves as a reverse strategy for identifying the appropriate binding pocket, thereby enhancing the reliability of the pocket assignments.
> >
> >
> > [1] Francoeur, Paul G., et al. "Three-dimensional convolutional neural networks and a cross-docked data set for structure-based drug design." Journal of chemical information and modeling 60.9 (2020): 4200-4215.
> >
> >
> > ## Regarding removing filtering steps for the test set to avoid the applicability of the benchmark being limited
> >
> > Thanks for your suggestion on the applicability of our work. For non-structure-based methods or models originally trained on datasets derived from different docking algorithms, these models can still be re-trained on our dataset to ensure a fair evaluation.
> >
> > We agree that the key to constructing a reliable machine learning task is to maintain consistency between the training and test datasets. However, it would then be "unfair" for other methods: like a "competition" our test set serves as a platform where all the models should compete within a same setting. as stated in our title, our primary goal is to define a new task for bioactivity prediction. **A task comprises the dataset, the label (prediction objective), and the split.** For model evaluation, this means models should be trained and evaluated on the dataset and learning objective we provide. **Our intention is not to evaluate a model trained on a different dataset with a different objective against our test set using our objective.**
> >
> > Additionally, the filtering process is for quality control to get a reliable dataset. As we discussed above, the consensus filter could improve the quality of the structural data we provided; thus, this quality control method is important and indispensible to our work in terms of data quality.

---

> > > ### Author Response · Authors · 2024-11-20
> > >
> > > ## Regarding the meaning of "unbias"
> > >
> > > Thanks for pointing out the unclear claim on the unbiased nature of our dataset. We try to give a more detailed explanation.
> > >
> > > The bias arises from the **different bioactivity ranges** across various protein pockets. For instance, pocket A may have an average bioactivity of 1.0, pocket B 5.0, and pocket C 10.0. Such bias is clearly shown in Figure 4(D).
> > >
> > > Consider a simple example with three pockets, each having five molecules with bioactivity labels: (0.8, 0.9, 1.0, 1.1, 1.2) for pocket A, (4.8, 4.9, 5.0, 5.1, 5.2) for pocket B, and (9.8, 9.9, 10.0, 10.1, 10.2) for pocket C. If the model only learns the range of each pocket and predicts the bioactivity **in reversed order** for the molecules—(1.2, 1.1, 1.0, 0.9, 0.8) for pocket A, (5.2, 5.1, 5.0, 4.9, 4.8) for pocket B, and (10.2, 10.1, 10.0, 9.9, 9.8) for pocket C—the overall Pearson correlation between predictions and ground truth labels would be greater than 0.99. However, this high correlation is misleading because it reflects the model's ability to learn the range bias of different pockets, not its ability to **correctly rank molecules within the same pocket.** This shows that while the model may seem accurate, it fails at the critical task of correctly ordering molecules by their bioactivity within each pocket.
> > >
> > > When calculating the correlation within each pocket, the model would show a correlation of -1, indicating that it fails to rank molecules correctly within the same pocket. This makes the model useless despite appearing effective when considering all pockets together.
> > >
> > > The "unbiased" nature of SIU stems from its approach of **evaluating a large number of molecules within a single protein pocket** and calculating correlations within the same pocket. This method ensures that bioactivity prediction metrics are reliable and free from inter-pocket biases. In contrast, previous benchmarking datasets like CASF-2016 and ATOM3D assess only one ligand per protein pocket and measure correlations across different pockets. As a result, many models may fail to accurately rank different molecules within a single pocket but still achieve good correlation metrics due to the inherent biases in those datasets.
> > >
> > > Thank you once again for highlighting the unclear justification regarding the term "unbiased." We will incorporate those explanations into our camera-ready version and make sure the concept is presented more clearly.
> > >
> > > ## Regarding the prediction for different label types
> > >
> > > Thank you very much for concerning the motivation for dividing predictions across different label types. We now understand and recognize that there may be coupling relationships between label types and proteins, which could lead to distributional differences across label types. To mitigate this effect, we have selected only pairs containing all four label types. That means for each label type, the corresponding list of proteins is the same. The new figure can be found here: https://anonymous.4open.science/r/SIU_ICLR-ADF1/new_violin.png
> > >
> > > Despite this refinement, the figures still indicate distribution differences across label types. Additionally, by ensuring that each pair includes all four types, we can calculate the Pearson correlation between different types for a given protein. The results are as follows:
> > >
> > > |       | IC50   | EC50   | Kd     | Ki     |
> > > |-------|--------|--------|--------|--------|
> > > | IC50  | 1      | 0.0773 | -0.0080 | 0.5849 |
> > > | EC50  | 0.0773 | 1      | 0.3752 | 0.2673 |
> > > | Kd    | -0.0080 | 0.3752 | 1      | 0.0463 |
> > > | Ki    | 0.5849 | 0.2673 | 0.0463 | 1      |
> > >
> > > Thus, we continue to believe that separating these types into distinct tasks is a worthwhile approach. Thank you again for prompting us to clarify our position.

---

> ### Author Response · Authors · 2024-11-20
>
> ## Regarding the term tradition/conventional approach
>
> Thanks for pointing this out. We realized that the terms the reviewer brought about are too genernal and might be misleading. We've added a determiner "machine learning" to these sentence to make it more clear.
>
> By predicting binding free energy, FEP+ might indeed be considered a bioactivity prediction method in some sense. However, the primary focus of our paper is on defining an unbiased machine learning bioactivity prediction task. The issue with FEP+ methods is that, although they might have high prediction accuracy, they can only process one target at a time, and there are also limitations regarding the small molecules that can be predicted (e.g. should have high similarity with one another). In one word, handling one pocket at a time should be considered as the inherent limitation of such methods, they are not actively doing such grouping with the intention of addressing biased problems when evaluating the model itself.
>
> Furthermore, when evaluating models, isn't it still necessary to calculate metrics as some average across different targets? Of course, if we report the metrics for each individual target, it would demonstrate that some models might perform better on specific targets. However, how do we compare the general performance of a model? This further brings up another issue with the FEP+ dataset: it includes too few targets, and for each target, there are very few small molecules. While this might work as a held-out test for machine learning models, these datasets address a completely different aspect from the motivation we aim to explore in our paper.
>
> # Response to "questions"
>
> ## Regarding the definition of success ratio
>
> The success ratio is defined as the generated pose has an RMSD < 2 with the ground truth ligand pose.
>
> ## Regarding the label types contained in our test set
>
> For the test set, we ensured that each label type used has a sufficient number of data points. Additionally, we treat different label types independently, meaning that Pearson and Spearman correlations are calculated separately for each label type. Additionally, as illustrated in the table in Appendix C, the number of small molecule-pocket pairs is substantial.
>
> ## Regarding the data availability
>
> The dataset will be released upon the acceptance of the paper. The license will be CC-BY-4.0. The released data includes:
>
> ### Split files
>
> Splits for the training and test sets of our task are provided. However, we do not impose a fixed split for the dataset, allowing users the flexibility to perform their own splits. The updated version of the dataset includes four different predefined splitting strategies:
>
> 1. 90% Sequence Identity Filter: A fixed test set is provided, and proteins with a sequence identity greater than 90% to the test set are removed from the training set.
> 2. 60% Sequence Identity Filter: A fixed test set is provided, and proteins with a sequence identity greater than 60% to the test set are removed from the training set.
> 3. 60% Sequence Identity + Structural Similarity Filter: A fixed test set is provided, and proteins with a sequence identity greater than 60% to the test set are removed from the training set. Additionally, protein pockets with a structural similarity greater than 20% to the test set are also excluded from the training set.
> 4. 10-Fold Cross-Validation Split: The dataset is divided into 10 clusters. Any pair of proteins with a sequence identity greater than 60% are placed within the same cluster. This split can be used for 10-fold cross-validation.
>
> These options provide users with the flexibility to evaluate models under various levels of sequence and structural similarity constraints.
>
> ### A pickle file that contains processed data and label for the dataset
>
> It is a dictionary that contains the processed information of the dataset. Each key is a uniprot id, and corresponding value is a list of dictionaries. Each dictionary is a data point and has following keys:
>
> {
>
> source data : PDB id and uniprot id information
>
> label : a dictionary for different labels, icluding ki, kd, ic50, ec50. Each label is a scalar value.
>
> ik : inchikey of molecule
>
> smi : smiles of molecule
>
> }
>
> ### structure files
>
> It is a dictionary of the following format:
>
> DIR
>
> ├── uniprotid
>
> │   ├── pdb id
>
> │   │   ├── pocket pdb file
>
> │   │   ├── inchikey1
>
> │   │   │   ├── pose1 sdf file
>
> │   │   │   ├── pose2 sdf file
>
> │   │   │   ├── pose3 sdf file
>
> │   │   ├── inchikey2
>
> │   │   │   ├── pose1 sdf file
>
> │   │   │   ├── pose2 sdf file
>
> │   │   │   ├── pose3 sdf file
>
> │   │   ...
>
> │   ...
>
>
> Users can use this to process their own data for their own models.

---

> > ### Author Response · Authors · 2024-11-20
> >
> > ## Regarding metric calculation details
> >
> > The task we focus on is a regression task aimed at predicting a scalar binding affinity value for each target-molecule pair. Pearson correlation is calculated across all available molecules, regardless of whether they are active or inactive. Our metric is designed to evaluate whether the ranking of predicted binding affinities aligns with the ranking of actual measured binding affinities. Defining "inactive" compounds can be challenging, as different criteria may be used across studies, making it difficult to provide an exact proportion of inactive molecules.
> >
> >
> > ## Regarding test set
> >
> > As detailed in Appendix C, we explain the process of constructing the test set. The selected targets represent a wide range of protein classes, including G-Protein Coupled Receptors (GPCRs), kinases, cytochromes, nuclear receptors, ion channels, epigenetic targets, and others, ensuring broad coverage of the bioactivity landscape. For instance, “C11B1\_HUMAN” belongs to the cytochrome P450 family, which plays a role in drug metabolism. “RARG\_HUMAN” is a nuclear receptor family member, with drugs like bexarotene used in certain cancers. “NMDE1\_HUMAN” represents the NMDA receptor, a key glutamate receptor implicated in neurological disorders, with memantine being an approved NMDA receptor antagonist for moderate to severe Alzheimer’s disease. Including these diverse targets enhances the relevance and applicability of our dataset for drug discovery.
> >
> > The curated test set consists of 10 protein targets, covering a diverse range of protein classes while maintaining an even distribution of small molecule-pocket pair counts. Each target contains approximately 2,000 pocket-molecule pairs, resulting in a total of more than 20,000 pairs. This exceeds the size of the entire PDBbind dataset, demonstrating the comprehensiveness of our test set.
> >
> > We also realize that it will be better to provide a larger test set. Thus e also provide a 10 folds version of our dataset. That means the dataset is divided into 10 clusters. Any pair of proteins with a sequence identity greater than 60% are placed within the same cluster. This split can be used for 10-fold cross-validation. The result for this dataset is shown in Appendix G table 6 in the updated paper.
> >
> >
> >
> > ## Regarding assay label types for grouping
> >
> > In our approach, we treat different label types separately. For each task, we focus exclusively on one specific label type. So you can say the label types are also grouped first.

---

> ### Comment · Reviewer_MuEF · 2024-11-25
> **Response to authors**
>
> Thank you for their detailed rebuttal, please try to include a summary of these explanations in the paper to improve its clarity. I recommend acceptance of this paper even in its current form but I will reiterate my recommendation to the authors to reconsider the way they construct the test set and provide the training data: the evaluation of bioactivity prediction task should not have to be dependent on a specific way of generating 3D structures, I expect many methods for this task that will arise in the coming years will not require a structural input and therefore this framing of the benchmark will severely limit its applicability.

---

> > ### Author Response · Authors · 2024-11-26
> >
> > Dear Reviewer,
> >
> > Thank you very much for taking the time to review our paper and rebuttals. We have updated the manuscript based on your valuable suggestions.
> >
> > Regarding cross-docked performance, we have included explanations and results in Appendix B.4 (Table 6). The term "assay type" has been updated to "label types" throughout the paper. A more detailed description of pocket definitions and deduplication has been added to Section 3.1, "Bioactivity label data cleaning and deduplication", and appendix C. Additionally, we have highlighted (in orange text) explanations of bias issues in previous evaluations and emphasized the unbiased nature of our task in the Introduction and Section 3.3. For label type distributions, we have added a corresponding figure in Appendix B.2, along with correlation metrics. Furthermore, we have included a discussion of the FEP dataset in Appendix B.6 and clarified the success ratio in the caption of Figure 3. A detailed description of the dataset is now provided in Appendix A. Details on mapping small molecules to pockets are included in Appendix C.1. We have also refined other small parts in the main text to improve clarity.
> >
> > Following your advice, we will provide a version of the test set where all pocket-molecule pairs are retained without applying a docking filter, as discussed in Section 3.2, "Versatile Usage". The corresponding training data, which includes non-structural small molecule information that can serve as inputs for methods not requiring structural data, will also be released alongside our structural and labeled datasets upon the acceptance of our paper. These updates are explicitly detailed in Sections 3.1 and 3.2 of the revised manuscript (highlighted in orange).
> >
> > We are truly grateful for your efforts and attention to detail, which have helped us refine the manuscript significantly. We hope these updates enhance the clarity and comprehensiveness of our paper and address any remaining ambiguities. Please do not hesitate to let us know if further modifications or additions are required.
> >
> > Your valuable and insightful feedback has been instrumental in shaping this work, and we thank you again for your time and dedication to improving the quality of our research.
> >
> > Best regards,
> >
> > The Authors

---

### Official Review · Reviewer_KtDx · 2024-11-02

**Soundness:** 2
**Presentation:** 2
**Contribution:** 3
**Rating:** 6
**Confidence:** 3

**Summary:**

The paper has two main contributions. First, it presents a new large-scale dataset of small molecule-protein interactions for unbiased bioactivity prediction, named SIU. Second, the authors redefine evaluation metrics for bioactivity prediction by proposing to average correlation coefficients per protein before calculating the overal coefficients, and by stratifying performance evaluation across four key types of bioactivity labels. The benefits of the proposed dataset and metrics are demonstrated empirically through the evaluation of several standard models.

**Strengths:**

Originality

The work’s originality lies primarily in its focus: rather than routinely proposing new models using existing datasets without analyzing their limitations, it critically examines the dataset and evaluation metrics themselves.

Quality

The technical quality of the work is high. The methods employed are appropriate and well-justified.

Clarity

The paper includes a thorough introduction that provides clear context and a strong foundation for the study.

Significance

This work is highly significant for the machine learning community, as it introduces both a new dataset and an updated evaluation protocol for the important task of bioactivity prediction, while effectively demonstrating limitations in previous datasets and evaluation methods.

**Weaknesses:**

Major Concerns:

- Data and Code Availability: A primary contribution of this work is the benchmark for bioactivity prediction, which includes a new dataset, evaluation metrics, and a specific data split. However, without readily accessible resources, the value of the work may be limited, as reproducing the benchmark independently could be challenging (e.g., requiring extensive runs across multiple docking software tools). It would be helpful if the authors addressed data availability, perhaps by providing a link to an anonymous GitHub repository.

- Simulated Poses vs. Crystal Structures: The dataset consists exclusively of simulated poses rather than crystal structures. Since the dataset is described as high-quality, a discussion of the tradeoffs between experimental crystal structures and generated poses would add valuable context. For example, the dataset is constructed based on a consensus from two out of three docking tools. Additional information on whether this approach assures high quality—especially if one tool generates a different pose—could be beneficial. The authors might also consider presenting results from training on SIU and evaluating on PDBbind to explore any potential distribution shifts.

- Sequence Identity-Based Data Splitting: The paper uses sequence identity-based data splitting; however, since the task is defined in terms of 3D structures, a structure-based split may offer a more appropriate approach, especially considering that similar structures may arise from different sequences.

- Validation Fold Construction: Details on the construction of the validation fold would clarify this important aspect, as it affects both model training outcomes and the usability of the dataset in future work.

- Test Fold Curation: While the manual curation of the test fold is appreciated, it would be helpful to understand the choice of focusing on only 10 protein targets, as this might affect the generalizability of the evaluation. Could the authors expand it to more proteins?

- Redefining the Bioactivity Prediction Task: Since the paper redefines the bioactivity prediction task, a clearer description of the specific inputs and outputs considered would be helpful. For instance, it is unclear which ligand pose is used as input when three docking tools are involved. Additionally, Section 3.3, "Reframing the Bioactivity Prediction Task," does not discuss the RMSE and MAE metrics used for evaluation.

- Figure 3 Interpretation: Interpreting Figure 3 is challenging without a more precise definition of the evaluation using co-crystal poses from PDB complexes. The text does not clarify which PDB complexes were included and does not clearly define Success Ratio and Remaining Ratio.

Minor Concerns:

- The abstract does not define what SIU stands for.
- The process for filtering based on extended-connectivity fingerprints could be elaborated. Is it based on pairwise Tanimoto similarities between fingerprints?
- Line 330: The statement, "Our structured approach facilitates nuanced assessments, such as evaluating the impact of specific small molecule modifications on protein interactions or comparing the efficacy of different compounds within the same protein pocket context," is not clear, especially as some filtering of ligands was involved based on extended-connectivity fingerprints.
- Line 282: The acronym "AIDD" could be defined for clarity.
- Line 346: It appears that Figure 4(D) should be replaced with Figure 4(B).

**Questions:**

- Could the authors clarify how the RMSE and MAE metrics are averaged? It would also be helpful to understand why RMSE* and MAE* are not defined using the same approach as Pearson* and Spearman* from Pearson and Spearman, respectively.
- Could the authors explain why grouping by pockets is considered equivalent to grouping by PDB IDs? Is not it possible for a single protein to have multiple pockets?

---

> ### Author Response · Authors · 2024-11-20
>
> Thank you very much for your valuable feedback and for recognizing our paper’s contribution in identifying issues with the previous task, as well as the significance of the SIU dataset and metrics. We are committed to addressing your concerns and questions to further clarify and improve our work.
>
> # Response to major concenrs
>
> ## Regarding Data availablity
>
> Our dataset contains a significant number of structure files and is quite large, making it impractical to host on an anonymous GitHub repository at this time. However, we plan to open-source the entire dataset upon acceptance, under the CC-BY-4.0 license. The dataset will include .pdb files for all protein target structures and .sdf files for all small molecule structures. Labels and metadata will be provided in pickle files, stored in dictionary format. Below is a detailed description of the dataset:
>
> ### split files
>
> Splits for the training and test sets of our task are provided. However, we do not impose a fixed split for the dataset, allowing users the flexibility to perform their own splits. The updated version of the dataset includes four different predefined splitting strategies:
>
> 1. 90% Sequence Identity Filter: A fixed test set is provided, and proteins with a sequence identity greater than 90% to the test set are removed from the training set.
> 2. 60% Sequence Identity Filter: A fixed test set is provided, and proteins with a sequence identity greater than 60% to the test set are removed from the training set.
> 3. 60% Sequence Identity + Structural Similarity Filter: A fixed test set is provided, and proteins with a sequence identity greater than 60% to the test set are removed from the training set. Additionally, protein pockets with a structural similarity greater than 20% to the test set are also excluded from the training set.
> 4. 10-Fold Cross-Validation Split: The dataset is divided into 10 clusters. Any pair of proteins with a sequence identity greater than 60% are placed within the same cluster. This split can be used for 10-fold cross-validation.
>
> These options provide users with the flexibility to evaluate models under various levels of sequence and structural similarity constraints.
>
> ### A pickle file that contains processed data and label for the dataset
>
> It is a dictionary that contains the processed information of the dataset. Each key is a UniProt ID, and corresponding value is a list of dictionaries. Each dictionary is a data point and has following keys:
>
> {
>
> source data : PDB ID and UniProt ID information
>
> label : a dictionary for different labels, including Ki, kd, ic50, ec50. Each label is a scalar value.
>
> ik : InChI key of molecule
>
> smi : SMILES of molecule
>
> }
>
> ### structure files
>
> It is a dictionary of the following format:
>
> DIR
>
> ├── uniprot id
>
> │   ├── pdb id
>
> │   │   ├── pocket pdb file
>
> │   │   ├── inchikey1
>
> │   │   │   ├── pose1 sdf file
>
> │   │   │   ├── pose2 sdf file
>
> │   │   │   ├── pose3 sdf file
>
> │   │   ├── inchikey2
>
> │   │   │   ├── pose1 sdf file
>
> │   │   │   ├── pose2 sdf file
>
> │   │   │   ├── pose3 sdf file
>
> │   │   ...
>
> │   ...
>
> Users can use this to process their own data for their own models.

---

> > ### Author Response · Authors · 2024-11-20
> >
> > ## Regarding Simulated Poses vs. Crystal Structures:
> >
> > Thank you for suggesting the inclusion of this discussion in our paper. We acknowledge that real crystal structures are typically of higher quality compared to simulated poses. However, the high cost of wet-lab experiments limits the availability of crystal structures, and many bioactivity measurements lack corresponding structural data. Our objective is to develop a large-scale dataset that provides multiple molecular structures and associated labels for each protein pocket, thereby redefining the structure-based bioactivity prediction task. We aim to strike a balance between quantity and quality by ensuring that the generated structures uphold a high standard of reliability.
> >
> >
> > And we've done extra experiments on benchmarking our consensus method on PDBbind dataset, comparing to crystal structures. We conducted this evaluation on the PDB refined dataset, which contains 5318 protein-small molecule complex structures. We compared the three docking softwares, and we calculated the average RMSD for poses that passed the voting filter and those that did not for each software. The results are shown in Table 1:
> >
> > ### Table 1: Comparison of Docking Software Using Voting Mechanism
> >
> > | Method | Avg RMSD (Passed Filter) | Avg RMSD (Failed Filter) | Percentage of Molecules Passing Filter |
> > |--------|--------------------------|--------------------------|-----------------------------------------|
> > | Glide  | 2.04                     | 5.08                    | 82.70%                                  |
> > | Gold   | 2.23                     | 6.00                    | 81.52%                                  |
> > | Vina   | 3.07                     | 5.81                    | 56.16%                                  |
> >
> >
> > From these results, it is evident that for each docking software, the molecules passing the filter exhibit a significantly lower average RMSD to the true pose compared to those that do not. This demonstrates the effectiveness of our method in ensuring the quality of generated poses.
> >
> >
> > ### distribution shifts
> >
> >
> > We also provide an experiment where the model is trained on the docked PDBbind structures and evaluated on our test set. The result is shown in Appendix F, and it demonstrates that the influence of distribution shift is minor.
> >
> > ## Regarding the data splitting
> >
> > Thanks for your valuable advice. Actually, as discussed above, we already removed structurally redundant pockets from the whole dataset. To further minimize the risk of information leakage via structural similarity, we adopted a more stringent filtering approach based on your suggestion. Using the FLAPP tool[1], we removed structurally similar pockets from the training set for the 0.6 sequence identity threshold, thereby creating a maximally out-of-distribution dataset. This setup allows for a rigorous evaluation of the model’s generalization capability. The results under this setting have been included in the updated appendix.
> >
> > Returning to the data splitting discussion: Unlike previous benchmarks that show artificially high results under loose splitting strategies but significant drops under more rigorous splits, our results demonstrate a different trend. Testing on our new unbiased metrics reveals that models perform poorly regardless of the splitting strategy, and their performance is not that sensitive to the data splitting approach. This suggests that the major concern is not overfitting due to data leakage, as was often the case in previous benchmarks.
> >
> > [1] Sankar, Santhosh, Naren Chandran Sakthivel, and Nagasuma Chandra. "Fast local alignment of protein pockets (FLAPP): a system-compiled program for large-scale binding site alignment." Journal of Chemical Information and Modeling 62.19 (2022): 4810-4819.

---

> ### Author Response · Authors · 2024-11-20
>
> ## Regarding the validation fold construction
>
> Currently, we use a random split to sample a portion of the training set to construct the validation set. This approach is based on our belief that defining the validation set is also a design choice left to the dataset users. For now, we focus on defining the standard train and test sets.
>
> However, following your advice, we have also provided a 10-fold version of our dataset. In this version, the dataset is divided into 10 clusters, where any pair of proteins with a sequence identity greater than 60% is placed in the same cluster. This split can be used for 10-fold cross-validation. The results for this dataset are presented in Appendix G, Table 6, of the updated paper.
>
>
> ## Regarding test set choice
>
> As detailed in Appendix C, we explain the process of constructing the test set. The selected targets represent a wide range of protein classes, including G-Protein Coupled Receptors (GPCRs), kinases, cytochromes, nuclear receptors, ion channels, epigenetic targets, and others, ensuring broad coverage of the bioactivity landscape. For instance, “C11B1_HUMAN” belongs to the cytochrome P450 family, which plays a role in drug metabolism. “RARG_HUMAN” is a nuclear receptor family member, with drugs like bexarotene used in certain cancers. “NMDE1_HUMAN” represents the NMDA receptor, a key glutamate receptor implicated in neurological disorders, with memantine being an approved NMDA receptor antagonist for moderate to severe Alzheimer’s disease. Including these diverse targets enhances the relevance and applicability of our dataset for drug discovery.
>
> The curated test set consists of 10 protein targets, covering a diverse range of protein classes while maintaining an even distribution of small molecule-pocket pair counts. Each target contains approximately 2,000 pocket-molecule pairs, resulting in a total of more than 20,000 pairs. This exceeds the size of the entire PDBbind dataset, demonstrating the comprehensiveness of our test set.
>
> Furthermore, generalization ability is not the primary concern for our task, as evidenced by the results. This is a challenging problem where models do not exhibit overfitting, reflecting the difficulty of the task.
>
> In response to your advice, we have also provided a 10-fold version of our dataset. In this version, the dataset is divided into 10 clusters, with any pair of proteins sharing a sequence identity greater than 60% placed in the same cluster. This split enables 10-fold cross-validation, and the results for this dataset are presented in Appendix G, Table 6, of the updated paper.
>
> ## Regarding Redefining the Bioactivity Prediction Task
>
> Thank you for pointing unclarity of input poses. it indeed highlights a challenge we faced during the construction of our pipeline. We established the selection priority as “Glide > GOLD > Vina” when a docking pose passed our consensus filter (i.e., two poses from different software with RMSD < 2 Å). All selected poses were saved in our final dataset and can be use as data augmentation during training.
>
> When utilizing the data, we recognized that these details also serve as justification for our methodology, and we have included this information in the appendix section for transparency and reproducibility.
>
> Regarding RMSE and MAE, we note that these metrics do not exhibit the bias issues associated with Pearson and Spearman correlation coefficients. Therefore, we retain the standard definitions for calculating RMSE and MAE in our evaluation.
>
>
> ## Regarding Figure 3 Interpretation
>
> We apologize for any confusion caused by Figure 3 and have updated the caption to provide greater clarity. Below is a detailed explanation of the figure:
>
>  •  Figure (a) illustrates an experiment designed to determine the optimal cutoff value for defining the consensus filter. The x-axis represents different cutoff RMSD values between poses generated by various docking methods. The success ratio is defined as the percentage of docking poses that pass the filter that can be aligned with the actual pose of the ligand. The remaining ratio indicates the percentage of docking poses that meet the filter criteria.
>
> •  Figures (B, C, and D) provide representative cases that demonstrate the differences in poses generated by various docking methods. These figures illustrate how poses from different methods compare and emphasize the role of the consensus filter in selecting reliable poses.
>
> We hope this detailed explanation resolves any confusion and makes the intent of Figure 3 clearer.

---

> > ### Author Response · Authors · 2024-11-20
> >
> > # Response to Minor Concerns
> >
> > ## Regarding abstract
> >
> > Thanks for the pointing this out, the corresponding sentence in the abstract has been changed to "To address these issues, we redefine the bioactivity prediction task by introducing the SIU dataset-a million-scale \textbf{S}tructural small molecule-protein \textbf{I}nteraction dataset for \textbf{U}nbiased bioactivity prediction task, which is 50 times larger than the widely used PDBbind."
> >
> > ## Regarding the process for filtering based on extended-connectivity fingerprints
> >
> > We apologize for previously placing this critical information in the Appendix. It has now been moved to the main text for clarity, and a more detailed explanation has been included.
> >
> > For small molecule deduplication, ECFP6 fingerprints were used to compare molecular structures. Molecules with high structural similarity (Tanimoto similarity > 0.8) were grouped into clusters. Representatives from each cluster, prioritized based on bioactivity, were selected to deduplicate the dataset while retaining structural diversity.
> >
> > ## Regarding the statement, "Our structured approach facilitates nuanced assessments, such as evaluating the impact of specific small molecule modifications on protein interactions or comparing the efficacy of different compounds within the same protein pocket context,"
> >
> > We understand that the reviewer might interpret "modifications" as exploring structure-activity relationships (SAR), which often involves small molecules with highly similar fragments. Such molecules might indeed be filtered out by our deduplication process.
> > Thank you for highlighting this point. Upon reflection, we realized that in this context, we were referring to transformation at the molecular level, rather than the substitution of small molecule fragments. We have revised the terminology to better clarify our intended meaning.
> >
> > Importantly, such deduplication was applied only to small molecules in pockets where the number of associated small molecules exceeded the 90th percentile (2,146) compared to that across all pockets. Additionally, as illustrated in the figure, while the total number of small molecules decreased, key properties—including molecular weight, oil-water partition coefficient (logP), hydrogen bond acceptor (HBA) count, and hydrogen bond donor (HBD) count—remained consistent, ensuring the essential characteristics of the dataset were preserved. The figure can be found at: https://anonymous.4open.science/r/SIU_ICLR-ADF1/diversity_stats.png
> >
> >
> > ## Regarding AIDD
> >
> > Thanks for pointing this out. We've added "AI-driven drug discovery (AIDD)" and checked all the abbreviations.
> >
> > ## Regarding figure 4
> >
> > Thank you for carefully reviewing the figure reference. However, we believe that "Figure 4(D)" is indeed the correct reference for line 346. In Figure 4, panels (A) and (B) illustrate differences across various label types, while panels (C) and (D) focus on differences across different protein targets. Line 346 belongs to the subsection "Unbiased correlation metrics with group-by-pocket approach," which pertains specifically to protein targets. Therefore, the appropriate reference here is "Figure 4(D)."
> >
> > We have updated the figure in the paper, and the updated figure can also be found at: https://anonymous.4open.science/r/SIU_ICLR-ADF1/distributions_update.png
> >
> >
> > # Response to Questions
> >
> > ## Regarding how the RMSE and MAE metrics are calculated.
> >
> > Regarding RMSE and MAE, we note that these metrics do not suffer from the bias issues associated with Pearson and Spearman correlation coefficients. As such, we retain the standard definitions for calculating RMSE and MAE in our evaluation, and they are not the core metrics we want to show or evaluate in our task. We just use them as reference.
> >
> > As discussed in our paper, the issue with how Pearson and Spearman correlations are typically calculated lies in the fact that they are computed across different protein pockets. This approach allows a model to achieve high correlation values simply by learning the affinity range or mean value for each pocket, without truly capturing the ability to rank different molecules within a single protein pocket. However, in drug discovery, the capability to accurately rank molecules for a specific protein pocket is far more critical, as it directly impacts the selection of potential drug candidates.
> >
> >
> >
> > ## Regarding grouping by pockets
> >
> >
> > We understand that the distinction between PDB ID and UniProt ID, as well as between pockets and protein targets, can sometimes be confusing. In our paper, we clarify that multiple pockets may arise from the same protein target, which is identified by its UniProt ID, not by its PDB ID. Actually, the pocket is defined as the surroundings of the corresponding co-crystal ligand. In this context, grouping by PDB ID is exactly grouping by pockets. We apologize for any confusion and have highlighted this distinction more clearly in the revised manuscript.

---

> > > ### Comment · Reviewer_KtDx · 2024-11-23
> > >
> > > I thank the authors for providing the clarifications to the manuscript. I am raising the score to 6 because, overall, the work provides a valuable contribution by building a new dataset based on public data and rethinking evaluation metrics for a biologically important task of bioactivity prediction, which is of a big importance for machine learning community. However, I cannot raise the score to 8 because some of the responses still leave room for improvement:
> > >
> > > - Regarding test set choice. It is still unclear why the authors have chosen only 10 protein targets for the test fold. In the response, the authors explain that these proteins are diverse and the corresponding number of pocket-molecule pairs exceeds the size of PDBbind. However, 10 proteins constitute only 0.1% of the collected dataset (to my understanding there are 9,544 in total in SIU), and I believe the number of test proteins could be increased. My main concern is that the evaluation on 10 proteins may be biased.
> > >
> > > - Regarding Figure 3 Interpretation. The authors do not seem to comment on which PDB complexes were used for the evaluation.
> > >
> > > - Appendix G appears to be empty. I guess it should contain a reference to Table 7.
> > >
> > > - It is not clear what method was evaluated for Table 7. Could the authors discuss the results from Table 7? It seems surprising, for example, that Pearson* on IC50 is the highest in the paper even though the FLAPP-based split should be the most challenging in the paper.
> > >
> > > - While an anonymous GitHub page is not critical, it would significantly strengthen my confidence that the presented work would be easy to use by machine learning audience, which is important for a dataset paper.
> > >
> > > Overall, I would suggest the authors improve the overall clarity and transparency of the work. Even though the work seems to be of high technical quality, it is hard to confidently assess it because of the missing explanations.

---

> ### Author Response · Authors · 2024-11-26
>
> Dear reviewer,
>
> Thank you very much for your detailed and insightful review of our paper and rebuttals. We greatly appreciate the time and effort you have dedicated to providing thoughtful feedback.
>
> ## Regarding test set choice
>
> Thank you for your valuable advice. The test set presented in the main text likely represents an experiment with biochemical generalizability, emphasizing targets of greater interest to our biochemical experts. This approach defines a biologically meaningful task that addresses previous limitations and issues in bioactivity prediction tasks. While we acknowledge its limitations, we have created a 10-fold version, allowing the model to be trained on 9 folds and tested on the remaining fold, which prioritizes using a setting that is more reliable for machine learning outcomes.
>
> The results from the 10-fold evaluation reemphasize our main conclusion that the proposed metrics are more challenging and that the current evaluation method is not appropriate, aligns with our primary goal to provide a more meaningful and unbiased bioactivity prediction task.
>
>
> ## Regarding Figure 3 Interpretation
>
> Thank you for your valuable comment. We apologize for the oversight regarding not including the PDB ID. We have now updated the caption of Figure 3 and also the Figure 3 itself in the revised version of the paper to include the PDB ID.
>
> In this section, our goal is to clearly demonstrate the effect of RMSD on the alignment of ligand docking poses, which also reflects the effectiveness of our structural data quality control process. To illustrate this, we randomly selected a PDB structure to visualize ligand poses docked by different docking software in our pipeline. The PDB ID used is 3PB7, which represents the crystal structure of the catalytic domain of human Golgi-resident glutaminyl cyclase. The small moleucle docked here is 6-[4-[(4-phenoxyphenyl)methyl]-1,2,4-triazol-3-yl]-1H-benzimidazole listed by both ChEMBL and BindingDB as an inhibitor of glutaminyl cyclase with a InChI key of CVGBPSGVCJCEFD-UHFFFAOYSA-N.
>
> ## Regarding adding a reference of Table 7 to Appendix G
>
> We sincerely thank you for taking the time to review the updated Appendix and for providing such insightful feedback. We deeply regret the oversight in its initial organization. In response to your suggestions, we have included the reference to all Tables , ensured all figures and tables are properly referenced and easily accessible, and thoroughly reorganized the Appendix to present the content in a more logical and coherent sequence. Additionally, we have included a list of contents to enhance readability and facilitate understanding.
>
> Your constructive comments have been instrumental in improving the quality of our submission, and we are truly grateful for your patience, guidance, and support.
>
> ## Regarding discussion of Table 7
>
> In Table 7 (currently Table 9), the results pertain to the Uni-Mol model.
>
> These results are actually comparable to those obtained using the original split version. A Pearson correlation within the range of 0.3 to 0.35 suggests a weak to moderate correlation, indicating the models are not accurate. This is noteworthy because, in the Atom3D LBA task, models with similar settings can achieve a Pearson correlation close to 0.8. This highlights the challenging nature of our task and suggests that the metrics are not inflated or overfitted, even when the data split is less strict. Consequently, it is reasonable that the metrics do not decline when the split becomes stricter.
>
> Additionally, it is notable that the conventional Pearson correlation (without grouping by pocket) for different label types drops obviously compared to the 0.6 version. However, the Pearson* metrics are not declined. This observation aligns with expectations: when correlations are calculated without grouping by PDB ID, the results are more susceptible to overfitting and potential data leakage. In contrast, for Pearson*, overfitting on the protein side does not necessarily lead to improved results.
>
> ## Regardinig providing an anonymous GitHub page
>
> Thank you for the suggestion. Our dataset exceeds 6 GB even when compressed as a ZIP file, which makes it challenging to host on an anonymized GitHub repository at this time. However, we can provide a version that includes example files.
>
> This is the example files of our dataset: https://anonymous.4open.science/r/SIU_ICLR-E619/example_files.zip
>
>
> ## summary
>
> Thank you once again for your insightful and constructive feedback! We truly appreciate your valuable input and thoughtful suggestions that help us refine our paper.
>
> Best regards,
>
> Authors

---

### Official Review · Reviewer_nZBp · 2024-11-02

**Soundness:** 3
**Presentation:** 3
**Contribution:** 3
**Rating:** 8
**Confidence:** 4

**Summary:**

The paper addresses significant issues in current evaluation strategies for bioactivity prediction, identifying problems with data preparation and evaluation metrics that result in inadequate benchmarking. For instance, approaches that only consider protein features in protein-small molecule interaction predictions may outperform methods that consider both due to overfitting. To tackle these challenges, the authors propose a new large dataset SIU based on mining wet lab bioactivity data and software-based docking to obtain structures, along with new evaluation metrics. The evaluations demonstrate the superiority of SIU over existing datasets and support the adoption of the new metrics.

**Strengths:**

**Originality**. The paper presents original research with novel isights into the issues in bioactivity predictions and novel methodology for constructing bioactivity data for machine learning.

**Quality**. The experiments are sound and convincing, clearly demonstrating the inadequacies of prior evaluation methods and the value of the new dataset and metrics.

**Clarity**. The paper is easy to follow.

**Significance**. The paper addresses a highly important issue of inadequate benchmarking in bioactivity prediction. The new SIU dataset is substantially larger than existing alternatives.

**Weaknesses:**

**Major Comments**

- The availability of the dataset is not discussed. Will it be publicly accessible, and if so, in what form?
- Data deduplication and diversity measurement methods are only briefly mentioned and not described anywhere in the text. Specifically, please elaborate on the ECFP and FLAPP methods used for deduplication of small molecules and pockets, respectively, and analyze the diversity and statistics after deduplication.
- Although the paper proposes a large dataset, further analysis of the data splitting approach would be beneficial. The meanings of 0.6 and 0.9 non-homology levels are unclear (Does 0.6 signify 60% sequence identity upon alignment? How are sequences aligned?). Furthermore, is sequence similarity-based splitting sufficient with respect to data leakage? Recent work suggests that structure-based splitting may be a better choice [[1](https://arxiv.org/abs/2402.18396), [2](https://www.biorxiv.org/content/10.1101/2024.07.17.603955v1)].
- The related work section is brief, only covering a few related datasets. Including related work on bioactivity prediction baselines to justify the selected methods for experiments and discussing prior evaluation metrics would strengthen the paper.
- A table summarizing the statistics (for example, number of proteins, ligands, pairs, unique pockets, unique ligands, etc.) for the training and test sets of SIU and other datasets like PDBbind would be beneficial.

**Minor Comments**

- Abstract: please clarify what SIU stands for. From the abstract it is not clear that SIU is a dataset.
- Figure 1: What is the GNN architecture?
- Figure 1C: the meaning of the red line and the target are not clear.
- Line 153: please specify the cutoffs being referenced.
- Figure 2: Poses are visualized in the same way as ligands, which is confusing.
- Line 209: “identified by PDB IDs” contradicts earlier text, which states that multiple pockets may arise from the same PDB ID.
- Figure 3: clarify the meaning of RMSD—is it the RMSD between a docking pose and the ground truth from PDB, or between docking poses obtained with different methods to quantify consensus?
- Section: “Structural data construction via multi-software docking”: Same as above, RMSD is used in two different senses, which is not always clear.
- Section: “Structural data construction via multi-software docking”: Are the three chosen docking methods independent? Why were these specific methods selected?
- Line 280: the claim “diverse small molecules” is unsupported by experiments.
- Line 346: “for each target” is slightly confusing—are there only 10 targets in the dataset?
- Line 392: clarify the statement, “We conducted non-homology analyses at two levels, 0.6 and 0.9”—does this imply a maximum of 60% and 90% sequence identity between training and test proteins?
- Line 424: UniMol reference appears to be missing.
- Tables 1 and 2: highlighting the highest values in bold would improve clarity.
- Line 496: “pdb” should be in uppercase (PDB).

**Questions:**

- Would a complementary analysis focused solely on small molecules, rather than only proteins, lead to the same conclusions? For example, could training a small-molecule-only baseline on PDBbind yield similar insights?

---

> ### Author Response · Authors · 2024-11-20
>
> Thank you very much for your valuable feedback and for recognizing our paper’s contribution in identifying issues with the previous task, as well as the significance of the SIU dataset and metrics. We are committed to addressing your concerns and questions to further clarify and improve our work.
>
>
> # Response to major concerns
>
> ## Regarding the dataset availability.
>
> We will open-source the whole dataset upon acceptance. The licence will be CC-BY-4.0. It will include the .pdb files for all the protein target structures and the .sdf files for all the small molecule structures. The labels and metadata are stored in pickle files in dictionary format. Here is a description of the dataset:
>
> ### split files
>
> Splits for the training and test sets of our task are provided. However, we do not impose a fixed split for the dataset, allowing users the flexibility to perform their own splits. The updated version of the dataset includes four different predefined splitting strategies:
>
> 1. 90% Sequence Identity Filter: A fixed test set is provided, and proteins with a sequence identity greater than 90% to the test set are removed from the training set.
> 2. 60% Sequence Identity Filter: A fixed test set is provided, and proteins with a sequence identity greater than 60% to the test set are removed from the training set.
> 3. 60% Sequence Identity + Structural Similarity Filter: A fixed test set is provided, and proteins with a sequence identity greater than 60% to the test set are removed from the training set. Additionally, protein pockets with a structural similarity greater than 20% to the test set are also excluded from the training set.
> 4. 10-Fold Cross-Validation Split: The dataset is divided into 10 clusters. Any pair of proteins with a sequence identity greater than 60% are placed within the same cluster. This split can be used for 10-fold cross-validation.
>
> These options provide users with the flexibility to evaluate models under various levels of sequence and structural similarity constraints.
>
> ### A pickle file that contains processed data and label for the dataset
>
> It is a dictionary that contains the processed information of the dataset. Each key is a UniProt ID, and corresponding value is a list of dictionaries. Each dictionary is a data point and has following keys:
>
> {
>
> source data : PDB ID and UniProt ID information
>
> label : a dictionary for different labels, including Ki, kd, ic50, ec50. Each label is a scalar value.
>
> ik : InChI key of molecule
>
> smi : SMILES of molecule
>
> }
>
> ### structure files
>
> It is a dictionary of the following format:
>
>
>
> DIR
>
> ├── uniprot id
>
> │   ├── pdb id
>
> │   │   ├── pocket pdb file
>
> │   │   ├── inchikey1
>
> │   │   │   ├── pose1 sdf file
>
> │   │   │   ├── pose2 sdf file
>
> │   │   │   ├── pose3 sdf file
>
> │   │   ├── inchikey2
>
> │   │   │   ├── pose1 sdf file
>
> │   │   │   ├── pose2 sdf file
>
> │   │   │   ├── pose3 sdf file
>
> │   │   ...
>
> │   ...
>
> Users can use this to process their own data for their own models.

---

> > ### Author Response · Authors · 2024-11-20
> >
> > ## Regarding deduplication and diversity
> >
> > We apologize for placing this important information in the Appendix section. To improve clarity, we have moved it to the main text and included a more detailed explanation.
> >
> > What should be highlighted here is that only the small molecules within pockets whose small molecule count exceeds the 90th percentile threshold (2146) across all pockets will be subjected to deduplication.
> >
> > For the deduplication of small molecules, Extended Connectivity Fingerprint (ECFP) [1] is used to compare molecular structures. ECFP is a type of molecular fingerprint used in cheminformatics to represent the structure of molecules. It is an extension of the circular fingerprint method, where a molecule is encoded into a fixed-length bit string that reflects its substructural features. ECFPs are widely used in various computational chemistry tasks, including virtual screening, similarity searching, and machine learning models. Molecules with high structural similarity (Tanimoto similarity > 0.8) are grouped into clusters. One or more representatives from each cluster, prioritized by bioactivity, are selected to deduplicate the dataset while retaining diverse structures.
> >
> > Fast Local Alignment of Protein Pockets (FLAPP) [2] is a program that used to calculate the structure similarity between two protein pockets. We remove structurally redundant pockets using this program.
> >
> > As illustrated in the figure: https://anonymous.4open.science/r/SIU_ICLR-ADF1/diversity_stats.png, the diversity of the small molecules is preserved. Key characteristics such as molecular weight, water-oil partition coefficient (logP), hydrogen bond acceptor (HBA) count, and hydrogen bond donor (HBD) count remain consistent, even though the total number of small molecules has been reduced.
> >
> >
> > [1] Rogers, David, and Mathew Hahn. "Extended-connectivity fingerprints." Journal of chemical information and modeling 50.5 (2010): 742-754.
> > [2] Sankar, Santhosh, Naren Chandran Sakthivel, and Nagasuma Chandra. "Fast local alignment of protein pockets (FLAPP): a system-compiled program for large-scale binding site alignment." Journal of Chemical Information and Modeling 62.19 (2022): 4810-4819.
> >
> >
> > ## Regarding the data splitting
> >
> > For the 0.6 and 0.9 non-homology levels, we set sequence identity thresholds at 60% and 90%, respectively. Sequence identity is calculated using alignment scores generated by the pairwise2.align function in BioPython. These thresholds are designed to test the model’s generalization ability across varying levels of sequence similarity.
> >
> > Actually, as discussed in previous section, we already removed structurally redundant pockets from the whole dataset. To further minimize the risk of information leakage via structural similarity, we adopted a more stringent filtering approach based on your suggestion. Using the FLAPP tool[1], we removed structurally similar pockets from the training set for the 0.6 sequence identity threshold, thereby creating a maximally out-of-distribution dataset. This setup allows for a rigorous evaluation of the model’s generalization capability. The results under this setting have been included in the updated appendix G.
> >
> > Returning to the data splitting discussion: Unlike previous benchmarks that show artificially high results under loose splitting strategies but significant drops under more rigorous splits, our results demonstrate a different trend. Testing on our new unbiased metrics reveals that models perform poorly regardless of the splitting strategy, and their performance is not that sensitive to the data splitting approach. This suggests that the major concern is not overfitting due to data leakage, as was often the case in previous benchmarks.
> >
> > [1] Sankar, Santhosh, Naren Chandran Sakthivel, and Nagasuma Chandra. "Fast local alignment of protein pockets (FLAPP): a system-compiled program for large-scale binding site alignment." Journal of Chemical Information and Modeling 62.19 (2022): 4810-4819.

---

> > > ### Author Response · Authors · 2024-11-20
> > >
> > > ## Regarding the related work
> > >
> > > We have updated the related work section in the paper. And this is the content:
> > >
> > >
> > > % \textcolor{blue}{\subsection{Bioactivity prediction models}}
> > > \textcolor{blue}{
> > > Atom3D also introduced two standard baseline models: a voxel-grid-based 3D convolutional neural network (3D-CNN) and a graph neural network (GNN) \citep{townshend2020atom3d}. Recent advances in bioactivity prediction have been driven by the application of pretrained models, such as Uni-Mol \citep{zhou2022uni} and ProFSA \citep{gao2023self}. These models utilize large-scale pretraining on molecular and structural data to achieve state-of-the-art performance on ATOM3D bioactivity prediction task. In Atom3D, binding affinity prediction models are evaluated using Root Mean Square Error (RMSE), Mean Absolute Error (MAE), Pearson correlation, and Spearman correlation metrics.}
> > >
> > > ## Regarding summarizing the statistics
> > >
> > >
> > > Thank you for pointing out that a table summarizing statistics can be beneficial. We have added this table to the appendix I. The table is also shown here:
> > >
> > >
> > > | Name     | Pocket-Molecule Pairs | Average Molecules per Pocket | Unique Pocket Number | Unique Molecule Number |
> > > |----------|-----------------------|-----------------------------|----------------------|------------------------|
> > > | PDBbind  | 19,443               | 1                           | 19,443              | 19,443                |
> > > | SIU      | 1,312,827            | 137.6                       | 9,544               | 214,686               |
> > >
> > > We also want to emphasize that a key contribution of our work, and the large-scale nature of our dataset, is the inclusion of multiple molecules per pocket.

---

> ### Author Response · Authors · 2024-11-20
>
> # Response to minor comments
>
> ## Regarding the Abstract
>
> Thanks for the suggestion, the corresponding sentence has been changed to "To address these issues, we redefine the bioactivity prediction task by introducing the SIU dataset-a million-scale \textbf{S}tructural small molecule-protein \textbf{I}nteraction dataset for \textbf{U}nbiased bioactivity prediction task, which is 50 times larger than the widely used PDBbind."
>
> ## GNN architecture
> The GNN model is adopted from the ATOM3D dataset[1]. Each atom is a node. It employs five layers of graph convolutions as defined by Kipf and Welling [2], with each layer followed by batch normalization and ReLU activation. The architecture concludes with two fully connected layers incorporating dropout.
>
> [1] Townshend, R. J., Vögele, M., Suriana, P., Derry, A., Powers, A., Laloudakis, Y., ... & Dror, R. O. (2020). Atom3d: Tasks on molecules in three dimensions. arXiv preprint arXiv:2012.04035.
>
> [2] Kipf, T. N., & Welling, M. (2016). Semi-supervised classification with graph convolutional networks. arXiv preprint arXiv:1609.02907.
>
> ## Unclarity in figure 1C
>
> Apologies for any confusion caused by the figure. The figure represents a single target and various molecules. The x-axis corresponds to the true label values of each molecule, sorted in ascending order, while the y-axis represents the predicted values for each molecule. The red line indicates y=x, which serves as the ideal reference line. Ideally, the blue dots should align closely with the red line. However, as shown in the figure, this is not the case. This suggests that the model is primarily predicting the mean value of the target for all molecules, rather than capturing the variance in the data.
>
> ## Regarding the cutoff
>
> It is updated in the paper.
>
> ## Regarding poses in Figure 2
>
> By "poses", we are specifically referring to ligand poses. We filtered the ligand poses generated by the docking software and stored the pocket structures and ligand poses separately to reduce storage costs. The pocket structures were not involved in the consensus filtering process, and the RMSD calculations were performed only between ligand poses generated by different software programs. However, the pocket structures are included in the final dataset.
> To address the confusion, we have redrawn Figure 2 for improved clarity. Specifically, we have updated the term "poses" to "ligand poses" and added the shapes representing pocket structures to indicate their inclusion in the final dataset. The updated figure can also be found at: https://anonymous.4open.science/r/SIU_ICLR-ADF1/update_data_process.png
>
> ## Regarding "identified by PDB IDs"
>
> Thanks for pointing this out. However, we have not made such a statement. We understand that the distinction between PDB ID and UniProt ID, as well as between pockets and protein targets, can sometimes be confusing. In our paper, we clarify that multiple pockets may arise from the same protein target, which is identified by its UniProt ID, not by its PDB ID. We apologize for any confusion and have highlighted this distinction more clearly in the revised paper.
>
> ## clarification on RMSD
>
> Thanks for pointing this out. In Figure 3, all RMSD refer to the the ones between poses generated by different docking softwares to quantify consensus.
>
> "A successful docking pose was defined as one with an RMSD of less than 2 Å compared to the experimental structure." This one is for the RMSD between the docking pose and ground truth PDB. Other RMSD are refered to the ones between poses generated by different docking softwares. We have updated the paper to make those more clear.

---

> > ### Author Response · Authors · 2024-11-20
> >
> > ## Regarding docking methods
> >
> > Yes, these three docking methods are independently developed by different companies or research groups, and they are based on distinct sampling algorithms. Glide, developed by Schrödinger, employs a systematic search-based sampling method. GOLD, from the Cambridge Crystallographic Data Centre (CCDC), uses a genetic algorithm for sampling. AutoDock Vina, created by the Center for Computational Structural Biology at The Scripps Research Institute, relies on an iterative local search approach.
> >
> > We chose these three software tools because they are among the most widely used and well-established in the field of computer-aided drug design (CADD). Furthermore, many influential studies have utilized or benchmarked these specific methods[1, 2].
> >
> > [1] Su, Minyi, et al. "Comparative assessment of scoring functions: the CASF-2016 update." Journal of chemical information and modeling 59.2 (2018): 895-913.
> >
> > [2] Bauer, Matthias R., et al. "Evaluation and optimization of virtual screening workflows with DEKOIS 2.0–a public library of challenging docking benchmark sets." Journal of chemical information and modeling 53.6 (2013): 1447-1462.
> >
> > ## Regarding diverse small molecules
> >
> > The diversity nature of our dataset lies in three ways:
> > (1) As the main motivation of our work is to address the biased issue in current bioactivity prediction tasks. The diversity of ligands corresponding to the same pocket is of great importance. This is self-explanary since PDBbind only have one co-crystal ligand structure within one pocket;
> > (2) Our diversity also lies in the large number of unique small molecules and unique protein targets we included. We've included an extra table to compare our SIU to thw main-stream dataset used in the bioactivity prediciton tasks to support this;
> > (3) As discussed above, the molecules in our SIU dataset show diverse range in terms of different properties.
> >
> > ## Regarding the "for each target"
> >
> > Thank you for pointing this out. There are 1720 targets in our SIU dataset.  We have rephrased the term to "for each of the representative targets" for clarity. Additionally, to provide av more comprehensive and convincing demonstration, we have included a heatmap and violin plot across more randomly sampled protein targets following the same logic in the Appendix. These representations further confirm that our findings remain consistent and valid in this broader context. The figure can also be found at: https://anonymous.4open.science/r/SIU_ICLR-ADF1/more_uniprots.png
> >
> >
> > ## Regarding non-homology analyses
> >
> > We apologize for any lack of clarity in our description. For the 0.6 and 0.9 non-homology levels, we set sequence identity thresholds at 60% and 90%, respectively. Specifically, the 0.6 non-homology level ensures that no sequences in the training and test sets share more than 60% sequence identity. Sequence identity is calculated based on alignment scores generated using the pairwise2.align function in BioPython.
> >
> > ## Regarding typos and citations
> >
> > Thank you for pointing them out. We have updated the paper to address those issues.
> >
> > # Reponse to the Question
> >
> > Thank you for your suggestion to conduct a complementary analysis focused solely on small molecules. We have completed this analysis using the same setting as the one used for the protein-only model, and the results are as follows:
> >
> > |               | Pearson | Spearman |
> > |---------------|---------|----------|
> > | Pocket+Mol    | 0.599   | 0.593    |
> > | Pocket Only   | 0.592   | 0.582    |
> > | Mol Only      | 0.491   | 0.479    |
> >
> > As shown in the table, using only molecular information does not achieve results comparable to models that incorporate both pocket and molecular information, or even those that rely solely on pocket information. This indicates that the challenge lies more on the protein side. These findings align with our motivation to define a more rigorous task aimed at testing a model’s ability to rank different molecules for a given protein.
> >
> > We also recognize that the task of ranking different protein pockets for a given molecule is an interesting and valuable direction. Designing a new dataset and task to address this problem is something we plan to explore in future work.

---

> > > ### Comment · Reviewer_nZBp · 2024-11-23
> > >
> > > Thank you for thoughtfully addressing my comments. After reviewing the authors’ responses to both my comments and those of other reviewers, I recommend the paper for acceptance and I raised my score accordingly. This work could be a valuable addition to the conference, as it addresses critical issues in current benchmarking practices for bioactivity prediction—specifically, the use of inappropriate metrics and flawed label preparation—and proposes a well-founded solution. The introduction of a new, large, high-quality dataset mined from public data, along with improved metrics, represents a valuable contribution to the field. However, I feel the paper could still benefit from greater clarity (e.g., the deduplication methodology remains somewhat difficult to understand) and putting the paper into a broader context (e.g., discussing implications for the trending field of protein-ligand docking, where, I believe, SIU may also be applicable). I believe that incorporating the detailed explanations provided in the authors’ responses into the final version of the paper, should it be accepted, would greatly enhance its overall quality.

---

> > > > ### Author Response · Authors · 2024-11-26
> > > >
> > > > Dear reviewer,
> > > >
> > > > Thank you for your recognition of our work and your thoughtful and constructive suggestions. We greatly appreciate the time and effort you have devoted to providing detailed feedback, which has been instrumental in improving our paper.
> > > >
> > > > We have carefully revised the manuscript in response to your comments, with particular attention to enhancing clarity. Specifically, we have addressed the deduplication methodology by adding more detailed explanations to make it clearer and easier to understand in section 3.1 as well as Apendix C.1.
> > > >
> > > > We also deeply appreciate your insightful recommendation to position our dataset and methods within a broader context. We are fully committed to ensuring this work is as impactful and beneficial as possible, not only for advancing bioactivity prediction but also for encouraging its adoption across related communities. To this end, in section 3.2, "Versatile-usage", and appendix F, we have elaborated on the potential applications of SIU in trending areas such as protein-ligand docking, virtual screening, and structure-based molecular generation, further highlighting its relevance and potential impact in these fields.
> > > >
> > > > Your recognition and thoughtful input have been incredibly encouraging, and we are sincerely grateful for your support. Should you have any additional suggestions or feedback, we would be more than happy to address them as we strive to make this work the best it can be. Thank you once again for your invaluable contributions.
> > > >
> > > >
> > > > Best regards,
> > > >
> > > > Authors

---

### Author Response · Authors · 2024-12-04
**Summary for the discussion period**

Dear AC and Reviewers,

We sincerely thank you for your thoughtful and constructive feedback, which has been instrumental in helping us refine and improve our work. We deeply appreciate the time and effort you have devoted to providing such detailed and valuable suggestions, which have made this journey a truly rewarding experience.

In this process, the reviewers' insights not only identified important writing improvements and additional experiments that we needed to add or improve, but also strengthened our confidence in proposing this work and its potential impact. **The positive comments can be summarized in the following aspects**:

- This work presents **valuable** (Reviewer 6s1k) and **novel** (Reviewer nZBp) insights, **identifying significant issues** (Reviewer nZBp and 6s1k) in the current bioactivity prediction task, which is a **critical task** (Reviewer nZBp and MuEF). This convincingly motivates our work (Reviewer 6s1k).

- We introduced a new, **large** (Reviewer nZBp, 6s1k, and KtDx), **high-quality** (Reviewer nZBp), and **carefully curated** (Reviewer 6s1k) dataset. We did a very extensive and carefully thought-out job in building this dataset (Reviewer MuEF) with novel methodology (Reviewer nZBp). The separation by different assay types is also useful (Reviewer 6s1k).

- The experiments are **sound and convincing**, clearly demonstrating the superiority of SIU over existing datasets and supporting the adoption of the new metrics (Reviewer nZBp). The technical quality of the work is high, and the methods employed are **appropriate and well-justified** (Reviewer KtDx).

- This work could be a valuable addition to the conference (Reviewer nZBp), representing **a significant contribution to the field** (Reviewer nZBp and MuEF). This work is **highly significant for the machine learning community** (Reviewer KtDx).


Throughout the review process, we have addressed most of the questions raised by the reviewers and followed their suggestions to improve our paper. In addition to responding to these concerns, we have taken great care to refine the visualizations, enhance the clarity of ambiguous descriptions, and provide practical details to improve the usability and accessibility of our work.

We have revised the paper by updating the details in Figures 2, 3, and 4, adding brief explanatory text for clarity. We also modified the captions for each figure, as per the reviewers' requests. All writing-related issues raised by the reviewers, including typos, abbreviations, and wording details, have been addressed. Furthermore, we have relocated important method details to the relevant sections of the main text and carefully revised them to ensure that all concerns raised by the reviewers about methods are thoroughly explained.

In line with the suggestions, we have expanded the description of our data-cleaning process, providing more granular details and including statistical presentations. We also added additional backing experiments for our multi-software docking and consensus filtering, particularly focusing on cross-docking and redocking, as suggested by the reviewers. Additionally, we evaluated the differences in assay values for various label types, considering potential coupling relationships between label types and proteins. This analysis yielded conclusions similar to our original findings, **further reinforcing our motivation**. We also analyzed assay value differences across a broader set of randomly sampled protein targets, which reinforced our initial analysis and further supported our approach. All of these additions have been included in the Appendix, as requested.

**Our dataset will be released upon acceptance of the paper**. In response to each reviewer's specific requests, we have incorporated the relevant dataset splitting settings and conducted experiments accordingly. The results from these experiments align with our initial settings and further demonstrate that our method is robust, with no risk of data leakage, and that our task is insensitive to dataset splitting.

Following the discussions and recognition from the reviewers, we are even more confident that our newly introduced dataset and task represent a significant contribution to the machine learning for drug discovery community. We hope they will inspire the development of more advanced models, ultimately enhancing the drug discovery process and driving meaningful progress in the field.

Thank you once again for your valuable guidance and for contributing to this rewarding collaborative process. It has been a privilege to learn from your expertise, and we are excited about the future contributions this work can make to the field.

Best regards,

Authors of #9500

---

### Meta-Review · Area_Chair_bwQ1 · 2024-12-19

**Metareview:**

In this paper, the authors propose a novel method for measuring bioactivity, aimed at improving predictions that better align with the objectives of assessing the activity of small molecules. More significantly, the paper introduces a new database that is 20 times larger than existing datasets and is set to be released alongside the paper.

Initially, the reviewers had numerous questions and concerns, but the authors addressed them comprehensively during the discussion phase, satisfying the reviewers and strengthening the paper's case for acceptance.

**Additional Comments On Reviewer Discussion:**

The reviewers and authors engaged positively, as mentioned in the metareview.

The paper can be spotlighted if there is a session for biology papers. Otherwise it can be presented as a poster.

---

### Decision · Program_Chairs · 2025-01-22

Accept (Poster)